# Angles Don't Lie: Unlocking Training-Efficient RL Through the Model's Own Signals

Qinsi Wang[1]    Jinghan Ke[2]    Hancheng Ye[1]    Yueqian Lin[1]    Yuzhe Fu[1]    Jianyi Zhang[1]
Kurt Keutzer[2]    Chenfeng Xu[2*]    Yiran Chen[1*]
[1]Duke University    [2]University of California, Berkeley

## Abstract

Current Reinforcement Fine-tuning (RFT) paradigms for Large Language Models (LLMs) suffer from sample inefficiency due to the redundant exposure of identical queries under uniform data sampling. While previous work has explored curriculum learning via heuristic difficulty metrics, these strategies exhibit limitations by neglecting the intrinsic learning signals generated by the model itself, thus leading to suboptimal training regimes. In this paper, we identify a model-inherent signal termed *angle concentration* that effectively reflects an LLM's capacity to learn from specific data. We theoretically and empirically demonstrate a correlation between the angular distribution of token hidden state vectors and the resulting gradient, revealing a learning preference for data exhibiting higher angle concentration. Inspired by this finding, we propose GAIN-RL, a Gradient-driven Angle-Informed Navigated RL framework. By leveraging the model's intrinsic angle concentration signal, GAIN-RL dynamically selects training data in each iteration, ensuring consistently impactful gradient updates and thus significantly enhancing overall training efficiency. Empirical evaluations show that GAIN-RL (GRPO) achieves over a $2.5\times$ acceleration in training efficiency across diverse mathematical and coding tasks and varying model scales. Furthermore, GAIN-RL (GRPO)'s efficient sampling yields data-efficient training, achieving better performance with half the original data compared to vanilla GRPO with full training data. Code is realsed at https://github.com/wangqinsi1/GAINRL/tree/main.

## 1 Introdction

Since the emergence of groundbreaking Reinforcement Learning Fine-tuning (RFT) techniques exemplified by Deepseek-R1 [1] and OpenAI's O1 [2], significant attention has converged on leveraging these approaches to enhance performance, notably in mathematical reasoning [3, 4, 5] and code generation tasks [6, 7]. This interest has catalyzed the development of numerous *algorithmic optimization methods*, including GRPO [8], ReMax [9], and Reinforce++ [10]. Despite such remarkable progress, critical challenges persist: RFT remains hindered by persistent issues of low sample efficiency and prohibitively high computational costs. For instance, the GRPO fine-tuning phase on Qwen 2.5-7B (Ray + vLLM) still consumed roughly 240 GPU hours (16 × H100-80 GB for 15h) to complete only 100 steps over 8k samples [11]. This low sample efficiency prompts the question: **Is it truly necessary to repeatedly expose every data point to the model hundreds of times?** Our answer is **No**. Humans adaptively learn by focusing on what they don't yet understand, rather than rote repetition of simple concepts. We extend this insight into reinforcement learning fine-tuning for LLM reasoning models from the perspective of data manipulation.

Manipulating the data is crucial for accelerating data-driven LLM training. Existing data manipulation techniques fall into two main categories: Sample selection methods, such as LIMO [12] and S1 [13],

---

*Corresponding authors.

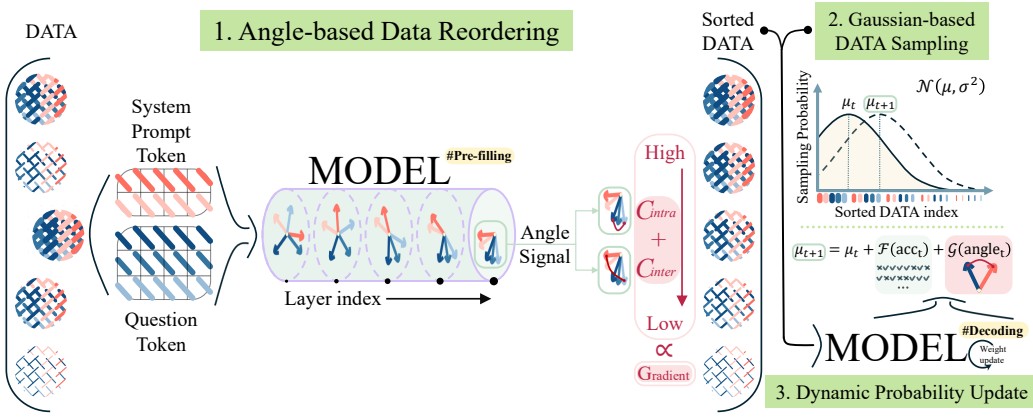

Figure 1: **Overview of GAIN-RL.** GAIN-RL consists of three steps: *(1)Angle-based Data Reordering*: Before training, the model pre-fills all data and ranks them by the combined angle concentration signals: $\mathcal{C}_{\text{inter}} + \mathcal{C}_{\text{intra}}$. *(2)Gaussian-based Data Sampling*: During training, each iteration begins by sampling reordered data using a Gaussian distribution. *(3)Dynamic Probability Update*: Epoch-wise accuracy and angle concentration are collected to dynamically update $\mu_{t+1}$. GAIN-RL guides the model to focus on high-angle, high-loss data, promoting effective gradients and faster convergence.

have shown that training on a carefully curated subset of high-quality data can improve performance. Separately, data ordering strategies, like ADARFT [14], have demonstrated that dynamically adjusting data difficulty during training can accelerate convergence. However, existing approaches suffer from two fundamental limitations that hinder their effectiveness in RFT. **First and foremost, current methods neglect the intrinsic characteristics of the models themselves.** They rely on fixed, model-agnostic criteria—such as difficulty or diversity—to evaluate data without accounting for how the target model itself perceives the data. We point out that different models interpret the same problem in markedly different ways. As shown in Fig. 2, different models produce diverging accuracy distributions on the same set of data, indicating that one-size-fits-all difficulty measures can lead to suboptimal training outcomes. **Second, existing methods suffer from high data preprocessing costs.** For instance, S1 and LIMO necessitate running large-scale models (e.g., Qwen2.5-Math-7B) across entire datasets to compute quality and difficulty scores. Similarly, curriculum learning often relies on manual annotation or expert-defined difficulty labels. These resource-intensive preprocessing steps significantly limit the scalability and practical responsiveness of these approaches.

We aim to attack the aforementioned challenges and propose *a model-informed signal* that (1) *reflects the learning capacity of a specific model on particular data*, (2) *incurs minimal computational costs*, and (3) *maintains generalizability across diverse models and datasets*. Achieving such requirements is inherently challenging, as accurately capturing model-data interactions typically needs a resource-intensive decoding stage. To overcome this challenge, we explore three key questions in this work:

1. **Which signal should we focus on?** By reformulating the gradient expressions, we find that the cosine similarity between token hidden states during inference (hereafter referred to as *angle concentration*) directly influences gradient norm, underscoring it as a critical signal.

2. **What are the characteristics of the signal?** By tracking the layer-wise evolution, we observe a transition from inter-segment to intra-segment angle concentration, which jointly facilitates information flow and constitutes the *Layer-wise Angle Concentration Pattern*.

3. **How can these signals accelerate training?** By monitoring angle concentration throughout training, we observe continuous convergence of intra- and inter-segment angles over epochs, revealing an *Epoch-wise Angle Concentration Pattern*. Moreover, the model preferentially learns samples with higher angle concentration before those with lower concentration, indicating a *Data-wise Angle Concentration Pattern*. These patterns highlight the model's data-learning preferences and can be leveraged to accelerate training.

Building upon these insights, we propose **GAIN-RL**, a **G**radient-driven **A**ngle-**I**nformed **N**avigated Reinforcement Learning Framework, illustrated in Fig. 1. GAIN-RL comprises three primary components: Data Reordering, Data Sampling and Probability Update. Before training, we reorder the training data based on model-informed angular concentration to enhance learning efficiency. During training, a dynamic Gaussian probability sampling strategy progressively guides the model towards data with lower angular concentration, with the pace adjusted according to real-time accuracy

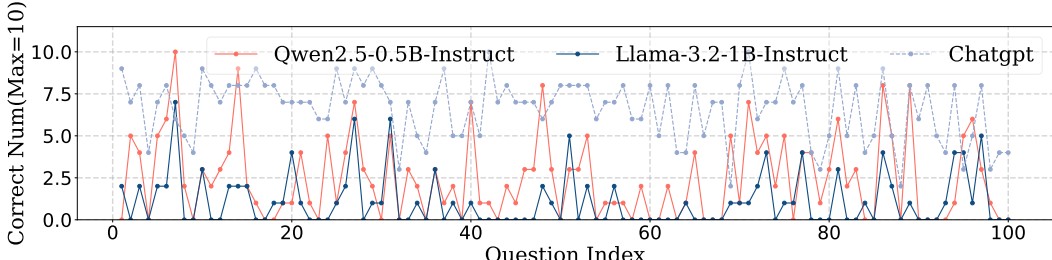

Figure 2: **Visualization of models' responses.** We evaluate the first 100 GSM8K questions by having Qwen2.5-0.5B-Instruct and LLaMA3.2-1B-Instruct each generate 10 answers per question, recording the number of correct responses. ChatGPT-4o rates question ease on a 0–10 scale (10 = easiest).

and angular signals. Notably, our data preprocessing involves only the inexpensive pre-filling stage, requiring under 10 minutes for over 7,000 samples. Overall, GAIN-RL provides a plug-and-play, broadly applicable, and data-efficient reinforcement learning solution. Experiments validate that GAIN-RL accelerates training by over 2.5× and, remarkably, surpasses full-sample performance using only half the data—demonstrating its powerful capability in driving data-efficient RFT.

In summary, our key contributions include:

- We propose that **the angle concentration serves as a critical signal** influencing gradient norm during training, thus predicting the model's ability to learn specific data samples.
- We demonstrate that **angle concentration intrinsically reflects information propagation during inference and learning dynamics during training,** and explicitly reveals Layer-wise, Epoch-wise, and Data-wise Angle Concentration Patterns.
- We introduce GAIN-RL, a framework that dynamically allocates training data per epoch based on angle concentration signals. **GAIN-RL is the first framework to utilize intrinsic model signals for data scheduling, providing a novel paradigm for efficient RFT.**

## 2   Model-Informed Data Evaluation Signals

With the goal of identifying signals that can effectively reflect a specific model's learning capability for data, in this section, we explore three key questions: (1) Which signal should we focus on? (2) What are the characteristics of this signal? (3) How can this signal be leveraged to accelerate training?

### 2.1   Which Signals Should We Focus On?

As mentioned in Sect. 1, obtaining model feedback via decoding is computationally expensive. In contrast, the pre-filling stage incurs significantly lower costs, as it requires only a single forward pass. Intuitively, we hope to identify signals from the pre-filling stage that can inform the training process.

Model training is driven by incremental gradient updates. Hence, to investigate how forward signals influence the backward process, we begin by examining and reformulating the gradient representation. Given a weight matrix $\boldsymbol{W} \in \mathbb{R}^{d \times h}$, the hidden states of input can be denoted as $\boldsymbol{x} \in \mathbb{R}^{m \times d}$, where $\boldsymbol{x}$ consists of $m$ tokens, and the hidden state of the $i$-th token is represented as $\boldsymbol{x}_i$. The output activation is denoted as $\boldsymbol{a} = \boldsymbol{x}\boldsymbol{W}$, where $\boldsymbol{a} \in \mathbb{R}^{m \times h}$. Assuming the loss function $\mathcal{L}$ has a gradient $\nabla_{\boldsymbol{a}}\mathcal{L} \in \mathbb{R}^{m \times h}$ with respect to the activation $\boldsymbol{a}$, the gradient of $\mathcal{L}$ with respect to $\boldsymbol{W}$ is given by

$$\nabla_{\boldsymbol{W}}\mathcal{L} = \boldsymbol{x}^\top \nabla_{\boldsymbol{a}}\mathcal{L} = \sum_{i=1}^{m} \boldsymbol{x}_i \left(\nabla_{\boldsymbol{a}}\mathcal{L}\right)_i, \tag{1}$$

where $\left(\nabla_{\boldsymbol{a}}\mathcal{L}\right)_i$ denotes the gradient of the loss $\mathcal{L}$ with respect to the $i$-th activation vector $\boldsymbol{a}_i$. To more precisely quantify the magnitude of the gradients, we consider the Frobenius norm of $\nabla_{\boldsymbol{W}}\mathcal{L}$. Leveraging the linearity of the Frobenius inner product, $\|\nabla_{\boldsymbol{W}}\mathcal{L}\|_F^2$ can be expanded as:

$$\|\nabla_{\boldsymbol{W}}\mathcal{L}\|_F^2 = \left\langle \sum_{i=1}^{m} \boldsymbol{x}_i \left(\nabla_{\boldsymbol{a}}\mathcal{L}\right)_i, \ \sum_{j=1}^{m} \boldsymbol{x}_j \left(\nabla_{\boldsymbol{a}}\mathcal{L}\right)_j \right\rangle_F = \sum_{i=1}^{m}\sum_{j=1}^{m} \left\langle \boldsymbol{x}_i \left(\nabla_{\boldsymbol{a}}\mathcal{L}\right)_i, \ \boldsymbol{x}_j \left(\nabla_{\boldsymbol{a}}\mathcal{L}\right)_j \right\rangle_F. \tag{2}$$

Next, to simplify Eq. 2, we utilize the compatibility between the Frobenius inner product and the matrix outer product. In particular, for any $\boldsymbol{u}, \boldsymbol{w} \in \mathbb{R}^d$ and $\boldsymbol{v}, \boldsymbol{z} \in \mathbb{R}^h$, the following identity holds:

$$\left\langle \boldsymbol{u}\,\boldsymbol{v}^\top, \ \boldsymbol{w}\,\boldsymbol{z}^\top \right\rangle_F = \text{tr}\!\left((\boldsymbol{u}\,\boldsymbol{v}^\top)^\top (\boldsymbol{w}\,\boldsymbol{z}^\top)\right) = \left(\boldsymbol{u}^\top \boldsymbol{w}\right)\left(\boldsymbol{v}^\top \boldsymbol{z}\right), \tag{3}$$

where $\mathrm{tr}$ denotes the matrix trace operator. Applying this identity term-wise in Eq. 2 gives

$$\|\nabla_{\boldsymbol{W}}\mathcal{L}\|_F^2 = \sum_{i=1}^{m}\sum_{j=1}^{m}(\boldsymbol{x}_i^\top \boldsymbol{x}_j)\big((\nabla_{\boldsymbol{a}}\mathcal{L})_i(\nabla_{\boldsymbol{a}}\mathcal{L})_j^\top\big) = \sum_{i=1}^{m}\sum_{j=1}^{m}\|\boldsymbol{x}_i\|\|\boldsymbol{x}_j\|\cos\theta_{i,j}\big((\nabla_{\boldsymbol{a}}\mathcal{L})_i(\nabla_{\boldsymbol{a}}\mathcal{L})_j^\top\big), \quad (4)$$

where $\cos\theta_{i,j}$ denotes the cosine similarity of the angle between $\boldsymbol{x}_i$ and $\boldsymbol{x}_j$. Eq. 4 explicitly reveals that, during inference, both the magnitudes of token hidden states and the angles between them directly influence the gradient values computed during backpropagation.

Since the magnitudes of token hidden states are normalized in each layer during inference, they cannot effectively convey useful information. Consequently, in this paper, we specifically focus on exploring the characteristics of relative angles of token hidden states. Furthermore, in the Appendix B.1, we provide proofs demonstrating that the nonlinear transformations in LLM inference—including both attention mechanisms and activation functions—are inherently angle-dependent and continuously modify the angles among token hidden states. We can now answer the first question:

**A1. We should focus on the relative angles between token hidden states during inference as it fundamentally impacts the gradient Frobenius norm computed during backpropagation. Specifically, the more concentrated the angles between tokens, the larger the gradient norm.**

## 2.2 What Are the Characteristics of This Signal?

In the previous subsection, we show that angles between token hidden states directly influence the gradients. To identify the characteristics we should focus on, we explore its layer-wise evolution.

**Layer-wise Observation.** We conducted experiments on the Qwen2.5-0.5b-Instruct model to observe the evolution process across different layers. The experimental results, shown as Fig. 3, clearly illustrate that in the initial layers of the model, no distinct pattern emerges, with the angles primarily determined by the input embeddings. As the layer depth increases, the angles gradually exhibit a segmented structure, whereby the hidden states of tokens within the same segment tend to cluster more closely. Upon further examination, we found these segments correspond precisely to distinct parts of the input sequence: the system prompt, few-shot examples, and the question. Eventually, in the final layers, angles between tokens from different segments begin to converge, reaching the highest degree of concentration. We give a more vivid demonstration in Fig. 1.

Based on the above observations, we introduce *Layer-wise Angle Concentration Pattern*: ***during inference, the model first induces intra-segment angle concentration and subsequently promotes inter-segment angle concentration.*** These two forms of clustering collaboratively facilitate information propagation through the model. A detailed demonstration is provided in Appendix C.1.

Therefore, assume the length of the input tokens is $m$. The first $n$ tokens constitute the system prompt and the few-shot examples, which are the same across all data samples. The remaining $m - n$ tokens represent the specific question to be answered. The characteristics we should focus on are:

$$\mathcal{C}_{\mathrm{intra}} = \frac{1}{(m-n)^2}\sum_{i=n+1}^{m}\sum_{j=n+1}^{m}\cos\theta_{i,j}, \quad \mathcal{C}_{\mathrm{inter}} = \frac{1}{(m-n)n}\sum_{i=n+1}^{m}\sum_{j=1}^{n}\cos\theta_{i,j}, \quad (5)$$

where $\mathcal{C}_{\mathrm{intra}}$ measures angle concentration within the question and $\mathcal{C}_{\mathrm{inter}}$ measures angle concentration between question tokens and the system prompt and few-shot tokens. We measure both at the final layer, where inter-segment clustering is maximal. Concentration within the system prompt and few-shot tokens is omitted, as it is constant across different questions.

**Attention-based Explanation.** To better understand the observed pattern, we also provide an analytical explanation from attention scores. In general, we find that *tokens with higher angle concentration correspond to higher attention scores*. Specifically, $\mathcal{C}_{\mathrm{intra}}$ represents the strength of attention within the question itself, while $\mathcal{C}_{\mathrm{inter}}$ indicates the model's ability to follow instructions. Furthermore, *the presence of sink attention encourages intra-segment and inter-segment angle concentration*. Details can be found in Appendix B.2. Our answer to the second question is:

**A2. The angles between token hidden states show both intra-segment and inter-segment concentrations during inference. In particular, we should pay attention to the final layer as the inter-segment clustering is most pronounced, resulting in the highest overall concentration.**

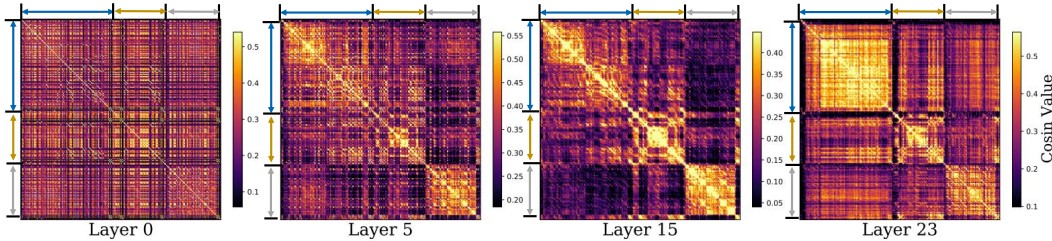

Figure 3: **Visualization of Layer-wise Concentration Pattern.** Experiment is performed on the Qwen2.5-0.5b-Instruct. In each subplot, the pixel at row i, column j represents the cosine similarity of the angle between the i-th and j-th token hidden states of the layer output. Blue, yellow, and gray arrows above the figure represent the tokens of the system prompt, few-shot examples and question, respectively. To better highlight the pattern, values are clipped between the 3rd and 97th percentiles.

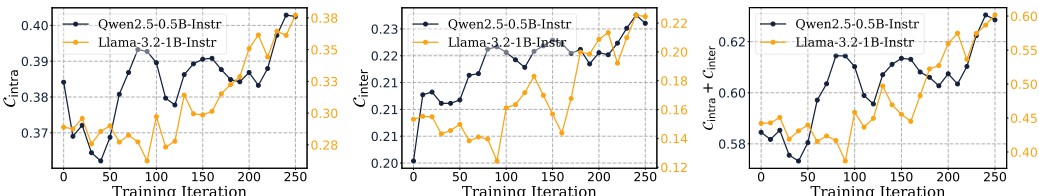

Figure 4: **Visualization of Epoch-wise Concentration Pattern.** (Left) $\mathcal{C}_{\text{intra}}$; (Mid) $\mathcal{C}_{\text{inter}}$; (Right) $\mathcal{C}_{\text{intra}} + \mathcal{C}_{\text{inter}}$. We train Qwen2.5-0.5B-Instruct and LLaMA3.2-1b-Instruct on GSM8K using GRPO for 250 iterations. To accelerate observation, we use a training batch size of 16, generate 4 responses per question, and set the learning rate to 1e-5. After each iteration, we perform pre-filling on the entire dataset and record the angle concentration from the hidden states of the final layer output.

### 2.3 How Can This Signal Be Leveraged to Accelerate Training?

Having established that intra-segment and inter-segment angular concentrations at the final layer are crucial characteristics, we now further examine their evolution during training to deepen our understanding of how they reflect and influence the training progress.

**Epoch-wise Observation.** To track how angular concentrations evolve during training, we perform inference on the same dataset on models of different epochs and monitor three signals: (1) Intra-question concentration $\mathcal{C}_{\text{intra}}$, (2) Inter-segment concentration $\mathcal{C}_{\text{inter}}$, (3) Combined signal $\mathcal{C}_{\text{intra}} + \mathcal{C}_{\text{inter}}$. As illustrated in Fig. 4, we can observe the *Epoch-wise Angle Concentration Pattern*: *during training, both inter-segment and intra-segment angle concentration increase progressively.* Additionally, we note that the intra-question concentration initially decreases before subsequently increasing, which suggests the model prioritizes mastering instruction-following capabilities before refining its focus internally on individual questions. These observations validate our hypotheses from the previous subsections, reinforcing that angular concentration effectively mirrors training dynamics.

**Data-wise Observation.** Furthermore, to examine data-wise angle behavior during training, we track the model's responses to samples with varying angle concentrations over epochs. As depicted in Fig. 5, surprisingly, the results revealed that *during training, the model tends to prioritize learning from higher-angle concentration data before addressing lower-angle concentration data,* which we introduce as *Data-wise Angle Concentration Pattern*. For instance, by iteration 100, questions with maximal angular measurements were almost entirely answered correctly, whereas those with smaller angles remained uncorrected. Note that the angle concentration distribution is measured on the untrained model, indicating that despite its evolution during training, the initial angle concentration of the data provides meaningful guidance for the training process.

To understand these patterns, we provide explanations from two perspectives: *gradients* and *neurons*.

**Gradient-based Explanation.** As shown in Eq. 4, gradients are influenced by both angle concentration and loss. Early in training, when losses are relatively uniform across samples, those with higher angle concentration receive stronger gradients and are learned faster. As training continues, angle concentration rises overall: high-angle samples, already mastered, see lower losses; while low-angle samples gain concentration, inherit larger gradients, and are learned next. This creates a natural, angle-driven learning progression. Unlike traditional curriculum learning based on task difficulty, angle concentration offers a more intuitive, model-centric training signal.

**Neuron-based Explanation.** Consider a FFN block with weights $\boldsymbol{W_u}$ and $\boldsymbol{W_d}$, and activation function SiLU. Given the input denoted as $\boldsymbol{x}$, the transformation process can be represented as:

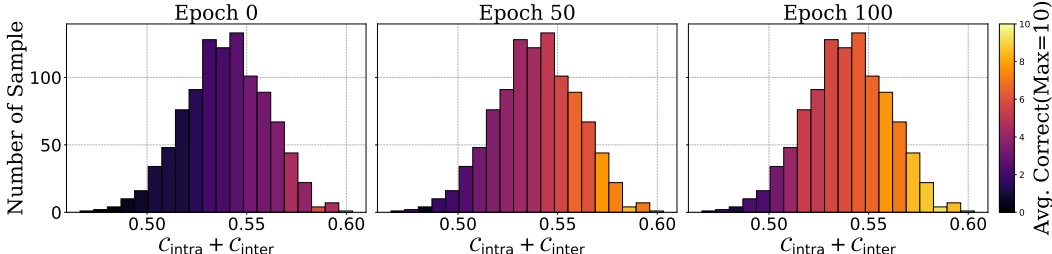

Figure 5: **Visualization of Data-wise Concentration Pattern.** Experiments are conducted on Qwen2.5-0.5b-Instruct. We first performed pre-filling on 1,000 samples of GSM8K using the untrained model to collect $\mathcal{C}_{\text{intra}} + \mathcal{C}_{\text{inter}}$ of samples, and plotted their statistical distributions (the histogram in the figure). Then we monitored the model responses to these samples at various iterations and recorded the average number of correct responses from samples in each angle concentration interval (brighter colors indicate a higher number of correctly answered samples within the interval).

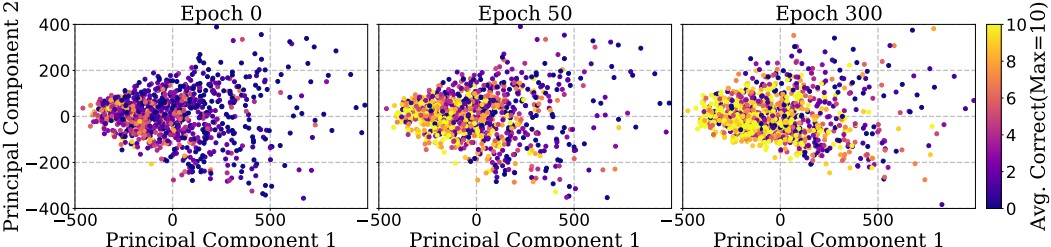

Figure 6: **Relationship between neuron activation patterns and accuracy over training.** At selected epochs, we collect both activation and answers for the first 1000 GSM8K samples. For each question, we identify the most frequently activated 20% of neurons across all tokens in the final layer, and use their indices to construct a binary core-neuron vector. We apply PCA to reduce these vectors to 2D. Each dot represents a sample; brighter colors indicate higher answer accuracy for the sample.

$z = xW_u$, $A = \text{SiLU}(z)$, $y = AW_d$, where $z$ and $A$ denote the pre-activation and activation outputs, respectively, and $y$ represents the output. The gradients of $j$-th neuron in $W_u$ and $W_d$ are

$$(\nabla_{W_u}\mathcal{L})_{:,j} = \sum_{i=1}^{m} x_i \left( (\nabla_A\mathcal{L})_{i,j} \, \text{SiLU}'(z_{i,j}) \right), (\nabla_{W_d}\mathcal{L})_{:,j} = \sum_{i=1}^{m} A_{i,j} (\nabla_y\mathcal{L})_i, \quad (6)$$

where $\text{SiLU}'(z_{i,j})$ is the derivative of the SiLU activation with respect to $z_{i,j}$. When $z_{i,j} < 0$, we have $\text{SiLU}'(z_{i,j}) \approx 0$ and $\text{SiLU}(z_{i,j}) \approx 0$. This indicates that the number of gradient components received by a neuron is proportional to the frequency of its activation.

Further analysis shows that tokens with higher angle concentration activate similar neurons due to shared value patterns (see Appendix B.3). These neurons receive stronger cumulative gradients, making them more effectively trained. Fig. 6 further illustrates how neuron activations converge during training, forming a distinct cluster correlated with higher accuracy. Samples activating neurons far from this cluster are harder to learn. Following prior work on neuron specialization, we hypothesize this cluster encodes domain-specific knowledge [15, 16, 17].

Synthesizing the above analyses, we provide an answer to the third question as follows:

**A3. We should follow the model's inherent learning dynamics — prioritizing higher-angle concentration data in the early stages and gradually transitioning to lower-angle concentration data. This progression ensures more effective gradient updates and improves training efficiency.**

## 3 GAIN-RL Framework

Based on the conclusions from the Sect. 2, we introduce **GAIN-RL**, a **G**radient-driven **A**ngle-**I**nformed **N**avigated-data **RL** framework, a plug-and-play training acceleration framework compatible with any model and dataset, incurring negligible costs. GAIN-RL consists of three components:

**Data Reordering Based on Angular Concentration.** Guided by our findings in Sect. 2.3—that models preferentially learn from data with higher angular concentration—we order the training data by angular concentration before training to improve efficiency. Given a model $M$ and a dataset $D = \{d_1, d_2, \ldots, d_N\}$, we first perform pre-filling on all data samples using $M$ to collect angular information. Subsequently, the data is sorted based on the combined signal at the final layer output,

$$\mathcal{C}_M(d_i) = \mathcal{C}^M_{\text{intra}}(d_i) + \mathcal{C}^M_{\text{inter}}(d_i), \ D_s = \text{Sort}_M(D; \mathcal{C}_M(d_i), \text{descending}) \quad (7)$$

Table 1: **Comparison of Pass@1 accuracy on Math benchmarks.** We report the accuracy at epoch 200 and the number of epochs needed to match vanilla GRPO's 200-epoch accuracy (Epo@Same Acc). ADARFT(GRPO) and GAIN-RL(GRPO) denote GRPO combined with the respective optimization.

| Experiments Setting | | Task Performance (200 Iteration) | | | | | | | Hardware Efficiency | |
| Model | Method | GSM8K [3] | Math [4] | AMC 23 [18] | AIME 24 [19] | Olympiad Bench[20] | Minerva Math[21] | Avg | Iter@ Same Acc | Speed Up |
| --- | --- | --- | --- | --- | --- | --- | --- | --- | --- | --- |
| Qwen 2.5 Math 1.5B Instruct | GRPO | 84.15 | 64.40 | 38.55 | 10.00 | 25.63 | 13.97 | 39.95 | 200 | 1× |
| | ADARFT(GRPO) | 85.52 | 66.00 | 40.96 | 13.33 | 26.07 | 14.71 | 41.09 | 150 | 1.33× |
| | **GAIN-RL(GRPO)** | **88.09** | **67.20** | **43.37** | **13.33** | **27.26** | **16.54** | **42.63** | **80** | **2.50×** |
| LLaMA 3.2 3B Instruct | GRPO | 74.60 | 40.20 | 19.28 | 6.67 | 11.70 | 8.46 | 26.8 | 200 | 1× |
| | ADARFT(GRPO) | 78.01 | 39.00 | 18.07 | 6.67 | 12.89 | 8.56 | 27.2 | 140 | 1.43× |
| | **GAIN-RL(GRPO)** | **76.04** | **42.00** | **21.69** | **6.67** | **14.22** | **10.29** | **28.5** | **80** | **2.50×** |
| Qwen 2.5 Math 7B Instruct | GRPO | 91.96 | 68.40 | 40.96 | 10.00 | 25.78 | 20.22 | 42.89 | 200 | 1× |
| | ADARFT(GRPO) | 92.65 | 70.20 | 42.17 | 10.00 | 25.33 | 21.32 | 43.61 | 150 | 1.33× |
| | **GAIN-RL(GRPO)** | **93.71** | **72.80** | **45.78** | **13.33** | **26.81** | **23.53** | **46.33** | **70** | **2.86×** |

Table 2: **Comparison of model performance on Code benchmarks.** ADARFT is not compared because DeepCoder lacks the difficulty coefficients required by ADARFT.

| Experiments Setting | | Task Performance (200 Iteration) | | | | | | Hardware Efficiency | |
| Model | Method | LCB [6] Pass@1 | LCB Pass@8 | Codeforces [22] Pass@1 | Codeforces Pass@8 | Humaneval+ [23] Pass@1 | Avg | Iter@ Same Acc | Speed Up |
| --- | --- | --- | --- | --- | --- | --- | --- | --- | --- |
| Qwen 2.5 Coder | GRPO | 10.8 | 21.5 | 5.15 | 17.8 | 78.3 | 26.7 | 200 | 1× |
| | **GAIN-RL(GRPO)** | **12.8** | **25.1** | **6.14** | **18.2** | **81.5** | **28.8** | **110** | **1.81×** |

the sorted dataset $D_s$ is directly employed in subsequent training. Notably, this sorting process is computationally efficient, as it only requires the pre-filling step, which can be efficiently batched. For example, sorting approximately 7000 samples of GSM8K with the Qwen-2.5-0.5-instruct model takes less than 10 minutes on a single NVIDIA A100 GPU. In contrast, previous approaches often required manual annotation or generation via large models, typically consuming several days.

**Data Sampling Guided by Gaussian Probability.** During training, we consistently prioritize data with higher angular concentration. At the $t^{th}$ training step, we assign sampling probabilities to each data sample in the sorted dataset $D_s = \{d_1^s, d_2^s, \ldots, d_N^s\}$ based on a Gaussian distribution parameterized by $\mu_t$ and $\sigma_t$. A subset $d^{(t)}$ of size $n$ is then sampled according to,

$$P_t(d_i^s) = \frac{1}{Z_t} \exp\left(-\frac{(i - \mu_t)^2}{2\sigma_t^2}\right), \quad d^{(t)} \sim \text{Sample}(D_s; P_t, n), \tag{8}$$

where $Z_t$ is a normalization constant ensuring that probabilities sum to unity. Employing probabilistic sampling instead of strictly sequential sampling enhances the stability and robustness of training.

**Probability Update Based on Accuracy and Angular Signals.** The Gaussian mean $\mu_t$ sets the peak-sampling region. We initialize $\mu_0 = 0$ to prioritize high-angle concentration data, then gradually increase $\mu_t$ as the model masters these samples, shifting focus to lower-angle concentration ones. At each step, $\mu_t$ is updated from the batch $d^{(t)}$ using its average accuracy and angle concentration,

$$\text{Acc}^{(t)} = \frac{1}{n} \sum_{i=1}^{n} \text{Acc}_{M_t}(d_i^{(t)}), \quad \mathcal{C}^{(t)} = \frac{1}{n} \sum_{i=1}^{n} \mathcal{C}_{M_t}(d_i^{(t)}), \tag{9}$$

where $M_t$ represents the model at the $t^{th}$ training step, and $\text{Acc}_{M_t}(d_i^{(t)})$ and $\mathcal{C}_{M_t}(d_i^{(t)})$ denote accuracy and angular concentration signals. Notably, computing these signals incurs no additional cost as model inference is inherently performed during training. The update rule for $\mu_{t+1}$ is given by,

$$\mu_{t+1} = \mu_t + \frac{n}{2} \cdot \tanh\left(\alpha(\text{Acc}^{(t)} - \beta)\right) + \frac{n}{2} \cdot \tanh\left(\gamma \cdot \mathcal{C}^{(t)}\right), \tag{10}$$

where $\alpha$ adjusts accuracy sensitivity, $\beta$ sets the target accuracy, and $\gamma$ controls angle sensitivity (see Sect. 4 for guidelines). This update strategy maintains high-gradient training by targeting samples near the desired accuracy while gradually incorporating harder, lower-angle data efficiently.

Appendix F experiments reveal that weighting the signal as $\mathcal{C}_{\text{intra}}^M + k \cdot \mathcal{C}_{\text{inter}}^M$ can further boost performance. For simplicity and broad applicability, we adopt the unweighted signal here and leave signal optimization to future work. Our main goal is to highlight angle concentration as a key signal.

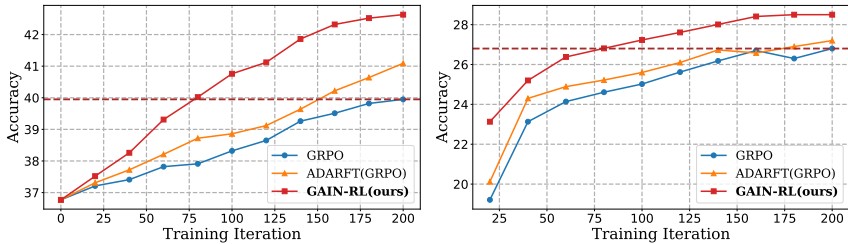

Figure 7: **Learning Dynamics of Different Methods on (Left) Qwen2.5-Math-1.5b-Instruct and (Right) LLaMA-3.2-3b-Instruct.** The y-axis shows average accuracy over GSM8K, Math, AMC 23, AIME 24, OlympiadBench, and Minerva Math. Performance is evaluated every 5 epochs.

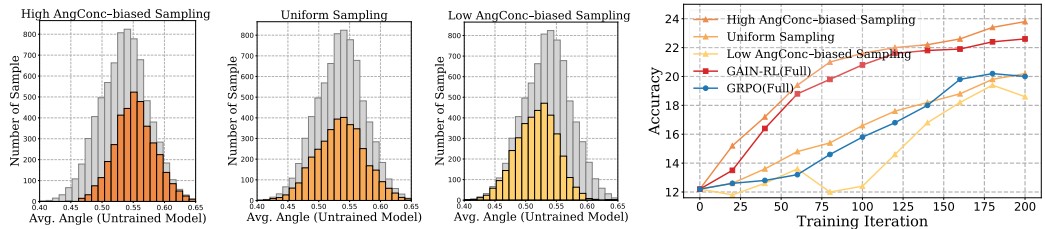

Figure 8: **Data Efficiency Analysis of GAIN-RL.** We sampled half of the data from the math dataset using three distinct sampling methods to train the Qwen2.5-0.5b-instruct model using GAIN-RL(GRPO): (1) High Angle Concentration-biased Sampling, (2) Uniform Sampling, and (3) Low Angle Concentration-biased Sampling. The first three figures illustrate the distribution of the sampled data (highlighted in bright colors) within the overall dataset (grey) under each sampling scenario. The rightmost figure presents the performance of models trained on differently distributed data, where GAIN-RL (Full) and GRPO (Full) denote results obtained by training with the complete dataset.

## 4 Experiments

To evaluate the effectiveness of GAIN-RL, we conducted a comprehensive experimental study on five levels: (1) Training Efficiency(Sect. 4.1), (2) Data Efficiency(Sect. 4.2), (3) RL Algorithms Generalization(Sect. 4.3), (4) Performance on Individual Tasks(Sect. 4.4), and (5) Ablation Studies(Sect. 4.5). For detailed descriptions of the used models and datasets, please refer to the Appendix E.

**Training and Hyperparameter Settings.** We set the target accuracy $\beta = 0.5$ to maintain strong gradients during training. Sensitivity parameters $\alpha = 2$ (for accuracy) and $\gamma = 0.5$ (for angle concentration) are tuned on a validation set to ensure stable learning, keeping the tanh function approximately linear over $\text{Acc}^{(t)} \in [0, 1]$ and $\mathcal{C}^{(t)} \in [-1, 1]$. Training is conducted using GRPO with a batch size and sampling number $n$ of 1024, implemented on the VerL framework with 8 NVIDIA A100 GPUs. Additional details are provided in the Appendix E.

**Baseline Settings.** We compare GAIN-RL with the vanilla GRPO and ADARFT[14], a state-of-the-art dynamic curriculum learning method in RL. The training settings are the same for all methods.

### 4.1 Training Efficiency of GAIN-RL

To evaluate the training efficiency of GAIN-RL, we use the DeepScaleR [24] and DeepCoder [25] datasets to train models for math and code tasks, respectively. They both cover diverse problem types and difficulty levels. Performance is tested on six math and three code benchmarks.

**Model Performance.** We report the performance of each method at 200 iterations during training. As demonstrated in Tab. 1 and Tab. 2, models trained with GAIN-RL(GRPO) exhibited superior performance across nearly all math and code datasets compared to models trained by vanilla GRPO and ADARFT(GRPO). Notably, with the Qwen2.5-Math-1.5B-Instruct model, GAIN-RL(GRPO) increased average accuracy across six math datasets from 39.95% and 41.09% to 42.63%, representing gains of 2.68% and 1.54%, respectively. Furthermore, the performance improvement observed on

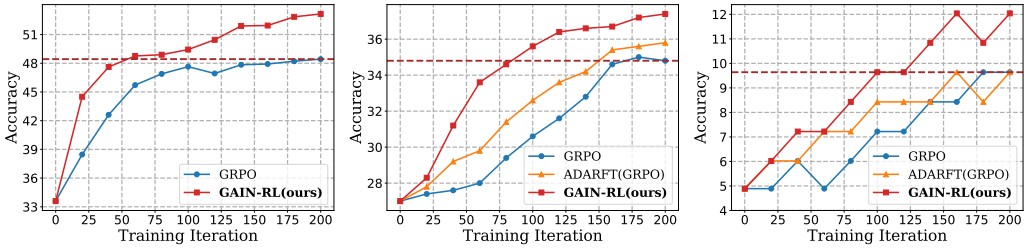

Figure 9: **Fine-tuning Performance on Single Tasks.** (Left) **GSM8k**. (Mid) **Math**. (Right) **AMC 23**. Models are trained on the training sets and evaluated on their validation sets. ADARFT is excluded from GSM8K due to missing difficulty coefficients.

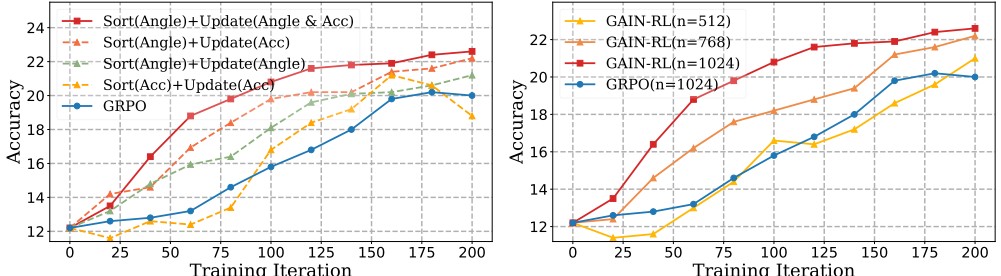

Figure 10: (Left) **Ablation Study of Module Components.** (Right) **Small Batch Scalability Test.** Experiments use Qwen2.5-0.5b-Instruct on Math training set, evaluate on the test set every 20 iterations.

LLaMA3.2-3B-Instruct demonstrates that GAIN-RL(GRPO) generalizes across different model families. These results highlight both model-level and task-level generality of our method.

**Hardware Efficiency.** To evaluate hardware efficiency, we report the number of iterations required for each method to reach the performance of vanilla GRPO at 200 iterations. As shown in Tab. 1 and Tab. 2, GAIN-RL(GRPO) requires approximately only half the iterations across various models and tasks. Specifically, for the Qwen2.5-Math-1.5B-Instruct and LLaMA3.2-3B-Instruct models, GAIN-RL(GRPO) achieves approximately 2.5× speedup, significantly outperforming ADARFT(GRPO) speedups of 1.33× and 1.43×, respectively. Fig. 7 further visualizes performance trajectories during training, where GAIN-RL(GRPO) consistently demonstrates faster convergence and superior performance at every iterations. As analyzed in Sect. 2, the acceleration mainly stems from the ability of our method to maintain strong gradient signals throughout training, leading to faster learning.

## 4.2 Data Efficiency of GAIN-RL

To further investigate whether GAIN-RL can enhance data effectiveness in RFT, we fine-tuned the Qwen2.5-0.5b-Instruct model on the Math dataset. We sampled half of the training dataset using three distinct distribution strategies for GAIN-RL(GRPO) training: (1) Uniform Sampling, where half of the training data was randomly selected; (2) High Angular Concentration-biased Sampling, where data points were sampled based on their angular concentration scores, assigning a linear sampling probability with a slope of 1 (prioritizing data points with higher angular concentration); and (3) Low Angular Concentration-biased Sampling, applying a linear sampling probability with a slope of -1.

## 4.3 RL Algorithms Generalization

To verify the algorithm generalization of GAIN-RL, we also combine GAIN-RL with PPO. We evaluate the final performance and hardware efficiency. Results in Tab. 3 highlight substantial gains in both performance and hardware efficiency with GAIN-RL(PPO), achieving an average of 2.2× training speedup across three mathematical benchmarks. This consistent benefit arises because gradient updates are fundamental across RL algorithms, validating GAIN-RL's universal acceleration capability.

Table 3: Performance of GAIN-RL combined with PPO. Qwen2.5-0.5b-Instruct is trained and evaluated on three datasets. Implementation details are in Appendix E.

| | Task Performance | | Hardware Efficiency | |
| --- | --- | --- | --- | --- |
| **Dataset** | **PPO** | **GAIN-RL (PPO)** | **Iter@ 200Acc** | **Speed Up** |
| GSM8K | 42.46 | **45.26** | 80 | **2.5×** |
| Math | 32.15 | **34.80** | 100 | **2.0×** |
| AMC 23 | 7.23 | **8.43** | 100 | **2.0×** |

We compared the performance of models trained under these sampling strategies. The experimental results, illustrated in Fig. 8, reveal that when using only half of the data, the model trained with High Angular Concentration-biased Sampling outperformed even the model trained with the full dataset. Based on the analysis in Sect. 2.3, we hypothesize this phenomenon occurs as data points with low angular concentration tend to produce smaller gradient updates and activate dispersed neural regions. Consequently, excluding these data points may enhance training efficiency. This insight provides new guidance for data selection: **prioritizing high angular concentration data and discarding low angular concentration data can significantly improve data effectiveness of RFT.** These findings highlight GAIN-RL's potential and effectiveness in guiding data selection. With uniform sampling, GAIN-RL using half the data performs slightly worse than with full data but remains on par with vanilla GRPO. In contrast, biased sampling toward low-angle concentration leads to unstable and poor results, consistent with our analysis of weaker gradients and dispersed neuron activations.

## 4.4 Performance on Individual Tasks

To further demonstrate the generality of GAIN-RL(GRPO), we also evaluate its performance on single-task RFT, which demands higher precision in data ordering and selection due to narrower difficulty ranges. Using Qwen2.5-0.5b-Instruct, we fine-tuned separately on GSM8K, MATH, and AMC training sets and evaluated on their test sets. Results in Fig. 9 indicate that GAIN-RL(GRPO) consistently yielded higher performance and efficiency. Specifically, on GSM8K, GAIN-RL(GRPO) achieved a $3.33\times$ training speedup and a $4.72\%$ final accuracy improvement. On the more challenging MATH and AMC 23 datasets, GAIN-RL(GRPO) also achieves $2.5\times$ and $2\times$ speedups, respectively. In contrast, ADARFT(GRPO) provided less improvement due to its fixed difficulty scoring, which may not align precisely with actual model-perceived difficulty. In contrast, by leveraging model-informed angle signals, GAIN-RL(GRPO) can predict learning priorities more accurately.

## 4.5 Ablation Studies

**Module Ablation.** To better understand the contribution of different components in GAIN-RL, we conduct ablation studies, including (1) Data ordering + Accuracy-based probability updates, (2) Data ordering + Angle-based probability updates, and (3) Accuracy-only (no ordering, discarding fully correct data to reduce training costs). As shown in Fig. 10 (left), all ablation variants exhibit degraded performance. The Accuracy-only group exhibits a marked performance drop, initially improving but later declining due to forgetting from prematurely discarding data. This underscores that data ordering and sampling constitute GAIN-RL(GRPO)'s primary performance advantages. Moreover, while both Data ordering + Accuracy-based probability updates and Data ordering + Angle-based probability updates show steady improvements over vanilla GRPO, they do not match the performance of GAIN-RL(GRPO), as each only captures a single aspect of gradient.

**Small-Batch Scalability Test.** To validate the scalability of GAIN-RL under reduced training batch sizes, we evaluated the model's performance with varying batch sizes. Specifically, we conducted experiments using the Qwen2.5-0.5b-instruct model on the Math dataset with training batch sizes of 512, 768, and 1024, respectively. The experimental outcomes, depicted in Fig. 10 (right), illustrate consistent and stable performance improvement across iterations for all batch sizes. Remarkably, even with the batch size reduced to half of the original size (n=512), which significantly reduces computational time and memory usage per iteration, GAIN-RL maintained performance comparable to vanilla GRPO. These results demonstrate that GAIN-RL effectively scales with smaller batch sizes, offering the flexibility to accelerate training by lowering batch size, with only minor performance degradation, especially beneficial under limited computational or memory resources.

## 5 Conclusion

We propose GAIN-RL, a novel Reinforcement Learning Fine-tuning framework that leverages the angle concentration signals to dynamically allocate training data at each iteration. GAIN-RL leverage the model's intrinsic angle concentration signal to dynamically selects training data in each iteration, ensuring consistently impactful gradient updates and thus significantly enhancing overall training efficiency. Furthermore, our empirical results further show that GAIN-RL (GRPO) achieves over a $2.5\times$ acceleration in training efficiency across diverse mathematical and coding tasks and varying model scales. Overall, GAIN-RL introduces a novel paradigm for training-efficiency RFT, highlighting how model-centric data-processing approaches can remedy the sub-optimality of current RFT methods and further elevate their effectiveness.

## Acknowledgements

Qinsi Wang, Jianyi Zhang and Yiran Chen disclose the support from NSF 2112562 and ARO W911NF-23-2-0224. We thank area chair and reviewers for their valuable comments.

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

**Organization** In this Appendix, we provide in-depth descriptions of the materials that are not covered in the main paper, and report additional experimental results. The document is organized as follows:

## A  Related Work

Reinforcement Fine-Tuning (RFT)[8, 10] has demonstrated significant effectiveness in enhancing the reasoning capabilities of large language models [15, 26, 27]. However, despite its promising potential, its low sample efficiency and high computational costs remain critical barriers to the broader adoption of RFT. Recent efforts aimed at addressing these efficiency have primarily focused on algorithmic optimizations and data-centric strategies. Algorithmic optimizations, exemplified by methods such as GRPO [8], REINFORCE++ [10], and ReMax [9], seek to reduce computational complexity by streamlining underlying RL algorithms. Although these methods typically improve efficiency and stability, they often involve inherent trade-offs. For instance, GRPO estimates advantages through relative comparisons within output groups, thereby eliminating the need for value functions. While this approach reduces complexity and the reliance on critics, it can introduce instability due to increased noise in advantage estimation, higher variance in updates, and greater sample requirements.

In parallel, data-centric strategies [28, 29, 30, 31, 32, 33, 34, 35, 36, 37, 38, 39, 40] have emerged as promising alternatives for efficient fine-tuning. For example, [28] provides the first quantitative audit of preference datasets, introducing metrics for scale, noise, and information density that expose quality bottlenecks before any policy update. Direct Preference Optimization (DPO) [41] simplifies the entire loop by replacing on-policy RL with a closed-form classification loss, making the choice of high-value preference data the primary driver of alignment quality. To further reduce annotation cost, Active Preference Optimization [42] casts RLHF as an active-learning bandit, adaptively querying only the prompts expected to maximize reward-model improvement. The latest data-centric strategies can be categorized based on their approach to data manipulation: data selection and data sequencing. Data selection techniques involve filtering extensive datasets to retain only a small subset of high-quality training data based on predefined metrics. Methods such as LIMO[12] and s1[13] have demonstrated that carefully curated small supervised fine-tuning datasets can achieve robust performance using orders of magnitude less data. On the other hand, data sequencing strategies[43] enhance model learning speed by rearranging the order of training data within existing datasets. Approaches like ADARFT[14] have shown that dynamically selecting data for each iteration can effectively accelerate the training process.

Nevertheless, existing data-centric strategies have generally failed to account for the unique characteristics of different models, applying uniform data handling procedures across diverse model architectures. Such uniformity can lead to suboptimal outcomes because models differ significantly in their sensitivity and response to the same datasets. To overcome this challenge, this paper aims to identify intrinsic signals within models that can reflect their perceptual capabilities toward data, thereby enabling tailored data strategies without incurring substantial additional costs.

# B Theoretical Supplement

In this section, we provide the theoretical explanation supporting the main text in Section 2. Specifically, we elaborate on the Angle-Dependent Effects of Attention and Activation (corresponding to Section 2.1 of the main text), the Attention-Based Explanation of Layer-wise Angle Concentration Patterns (corresponding to Section 2.2 of the main text), and the Neuron-Based Explanation of Data-wise Angle Concentration Patterns (corresponding to Section 2.3 of the main text).

## B.1 Theoretical Justification of Angle-Dependent Effects in LLMs

In this section, we demonstrate through derivation of the LLM computational process that the nonlinear operations in LLMs—namely attention and activation computations—are inherently dependent on the angles between the input hidden states.

To support this claim, we first introduce two empirical assumptions based on observation:

_Observation 1._ $W_q$ _and_ $W_k$ _are nearly approximately orthogonal to each other, i.e.,_ $W_q W_k^T \approx \theta I$. $\theta$ _is a constant._ $\left(W_o, W_u, W_d\right)$ _are approximately orthogonal matrices, i.e.,_ $W W^T \approx \lambda I$. $\lambda$ _is a constant._

_Observation 2._ _For activation function output vectors_ $A_i$ _and_ $A_M$, $cos(\angle(A_i, A_M))$ _is proportional to the number of intersections of activated neurons, i.e.,_ $\left|\Gamma(x_i) \cap \Gamma(x_M)\right|$.

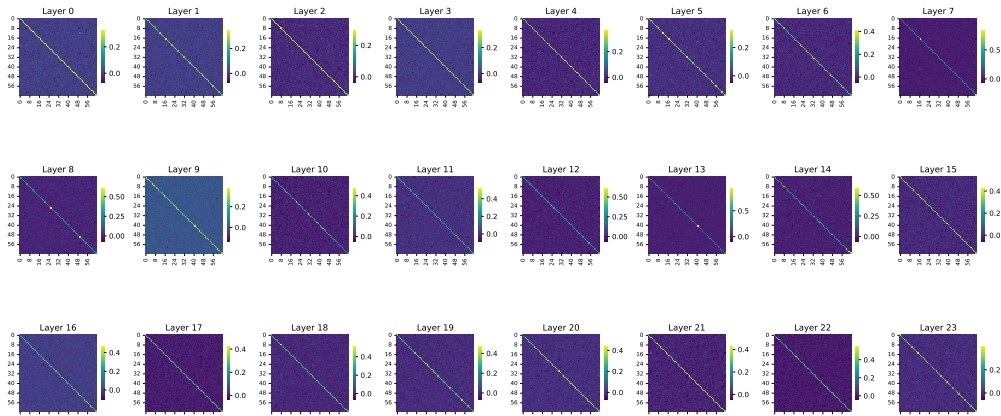

Figure 11: Visualization of $W_D @ W_D.T$ at different layers in Qwen2.5-0.5B-Instruct.

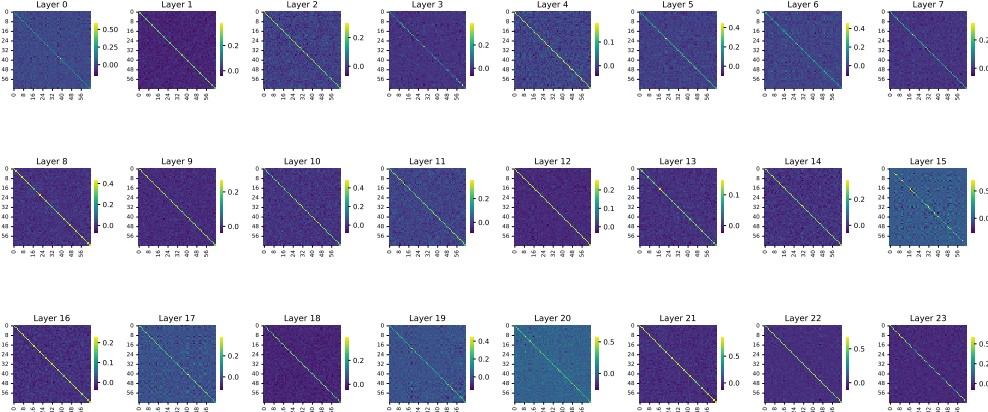

Figure 12: Visualization of $W_O @ W_O.T$ at different layers in Qwen2.5-0.5B-Instruct.

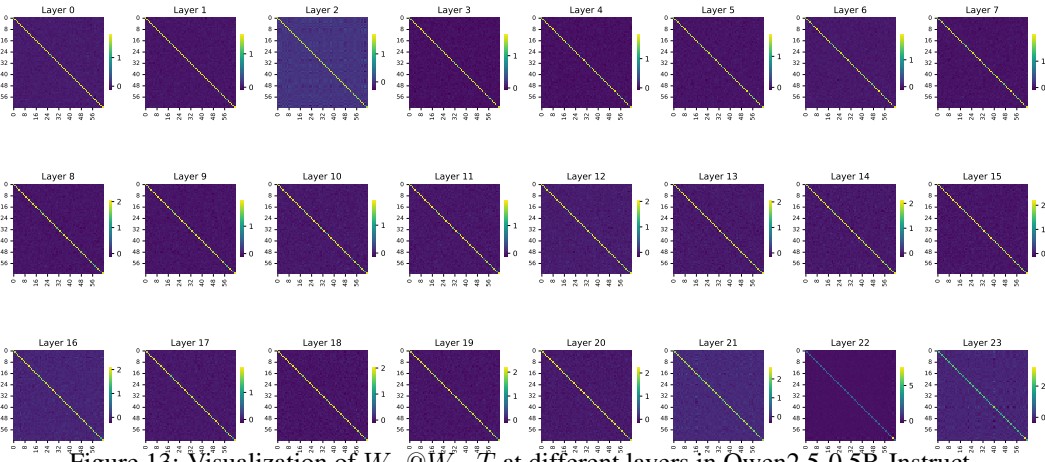

Figure 13: Visualization of $W_U @ W_U.T$ at different layers in Qwen2.5-0.5B-Instruct.

We provide empirical validation to support both of these assumptions.

For **Observation 1**, we show $W_Q @ W_K.T$, $W_V @ W_V.T$, $W_D @ W_D.T$ of all different layers of Qwen2.5-0.5B-Instruct in Fig. 11, 12, and 13. It can be seen that different layers have this orthogonal relationship. In fact, the orthogonal relationship of matrices in neural networks has been studied since a long time ago. In particular, [44] proposed a new regularization method that encourages the weight matrix of the neural network to maintain orthogonality during training by introducing a self-orthogonality module. This method helps to improve the training stability and generalization ability of the model. [45, 46, 47] explores adding orthogonal regularization to weights during training to improve training stability. The author proposed an orthogonal regularization method for weights, aiming to solve the gradient vanishing and explosion problems encountered by deep convolutional neural networks during training. It can be seen that modules with orthogonality are found in various different models to improve the training stability and performance of the model. To the best of our knowledge, we are the first work to intuitively show this orthogonal performance in LLM, which can be more fully explored in subsequent research.

For **Observation 2**, this observation is illustrated in Fig. 14. An intuitive understanding is that if $x_i$ and $x_M$ activate more of the same neurons, $A_i$ and $A_M$ will have more positive values in common positions, making $\cos(A_i, A_M)$ larger.

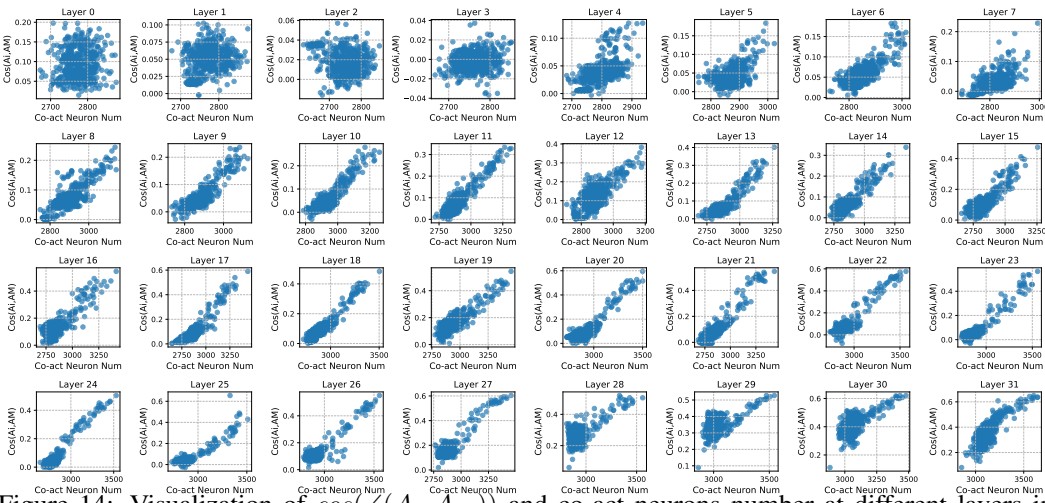

Figure 14: Visualization of $cos(\angle(A_i, A_M))$ and co-act neurons number at different layers in Qwen2.5-0.5B-Instruct.

_Theoretical Insight 1. Within an attention block, the degree of interaction between two tokens is governed by the relative angle of their input hidden states._

*Justification.* For a single token, suppose its input to the Attention block is $y$. Consider a sequence of tokens $[y_1, y_2, \ldots, y_M]$, for token $y_m$, its computation in the Attention layer can be expressed as:

$$\hat{y}_m = \text{LayerNorm}(y_m), \quad V_m = \hat{y}_m \, W_v,$$
$$\alpha_{im} = Softmax\big((\hat{y}_i \, W_q) \, (\hat{y}_m \, W_k)^T\big), \quad i < m \tag{11}$$
$$O_m = \alpha_{1m} \, V_1 + \alpha_{2m} \, V_2 + \cdots + \alpha_{mm} \, V_m,$$

where $\alpha_{im}$ is the attention score between the $i$-th and the $m$-th token. $O_m$ is the output vector of the $m$-th token. To examine the influence of the $i$-th token $y_i$ on the final output $O_m$ of the $m$-th token, we can consider the following projection value:

$$\big\|\text{Proj}_{O_m}\big(\alpha_{im} \, V_i\big)\big\| = \|\alpha_{im} \, V_i\| \cos\big(\angle(V_i, O_m)\big). \tag{12}$$

where $\big\|\text{Proj}_{O_m}\big(\alpha_{im} \, V_i\big)\big\|$ is the projection value of $\alpha_{im} \, V_i$ on $O_i$. And $\cos\big(\angle(V_i, O_m)\big)$ is the cosine value of the angle between the $V_i$ and $O_m$ vectors. From Eq. 11, $O_m$ is a sum of vectors in different directions. Since the self-attention score $\alpha_{mm}$ is typically much higher than $\alpha_{im}$ for other tokens, we can simplify the projection by assuming that $O_m$ is primarily determined by $\alpha_{mm} V_m$, i.e.,

$$\big\|\text{Proj}_{O_m}(\alpha_{im} V_i)\big\| \approx \|\alpha_{im} V_i\| \cos(\angle(V_i, V_m)), \tag{13}$$

where $\alpha_{im} = Softmax\big((\hat{y}_i \, W_q) \, (\hat{y}_m \, W_k)^T / \sqrt{d}\big)$. Since the Softmax function is monotonic, that is, $Softmax(x) \propto x$. We can have $\alpha_{im} \propto (\hat{y}_i \, W_q) \, (\hat{y}_m \, W_k)^T$. Therefore,

$$\|\alpha_{im} V_i\| \cos(\angle(V_i, V_m))$$
$$\propto (\hat{y}_i \, W_q) \, (\hat{y}_m \, W_k)^T \|V_i\| \cos(\angle(V_i, V_m))$$
$$= \langle \hat{y}_i W_q, \hat{y}_m W_k \rangle \langle V_i, V_m \rangle / \|V_m\| \tag{14}$$
$$= \big(\hat{y}_i (W_q W_k^T) \hat{y}_m^T\big)\big(\hat{y}_i (W_v W_v^T) \hat{y}_M^T\big) / \|V_m\|$$

Based on the Observation 1, $W_q W_k^T \approx \theta I$, $W_v W_v^T \approx \lambda I$. Therefore, combining Eq. 13 and Eq. 14,

$$\big\|\text{Proj}_{O_m}(\alpha_{im} V_i)\big\| \propto \theta \lambda \langle \hat{y}_i, \hat{y}_m \rangle \langle \hat{y}_i, \hat{y}_m \rangle / \|V_m\| \tag{15}$$

Given that $\hat{y}$ is the result of $y$ after LayerNorm, $\hat{y}$ and $y$ share the same direction, and $\| \hat{y} \| = 1$, it holds that $\langle \hat{y}_i, \hat{y_m} \rangle = cos(\angle(y_i, y_m))$. We can have

$$\big\|\text{Proj}_{O_m}(\alpha_{im} V_i)\big\| \propto \cos(\angle(y_i, y_m)), \tag{16}$$

Eq. 19 shows that the angle between different tokens directly affects their mutual interaction in attention layer. The closer the angles of two tokens, the greater their mutual influence. $\qquad \square$

We also demonstrate that the computation of activations is influenced by the angular relationships between hidden states. This justification is presented as Theoretical Insight 4 in Section B.3.

## B.2 Attention-Based Explanation of Layer-wise Angle Concentration Patterns

In this section, we present an attention-based explanation of Layer-wise angular concentration patterns. Specifically, we theoretically show that: (1) the degree of angular concentration between two hidden states influences the magnitude of their attention scores, and (2) the presence of sink attention structure encourages angular concentration among tokens.

*Theoretical Insight 2. The smaller the angle between the input hidden states of two tokens to the attention block, the higher their corresponding attention score.*

*Justification.* For two tokens $i$ and $j$, let their inputs to an attention block be denoted as $x_i$ and $x_j$, respectively. Then, their attention score $\alpha_{ij}$ can be expressed as:

$$\hat{y}_i = \text{LayerNorm}(y_i), \quad \hat{y}_j = \text{LayerNorm}(y_j),$$
$$\alpha_{ij} = Softmax\big((\hat{y}_i \, W_q) \, (\hat{y}_j \, W_k)^T / \sqrt{d}\big), \tag{17}$$

Based on the Observation 1, $W_q W_k^T \approx \theta I$, Therefore,

$$\alpha_{ij} = Softmax\big((\hat{y}_i \, W_q) \, (\hat{y}_j \, W_k)^T / \sqrt{d}\big) = Softmax\big(\langle \hat{y}_i W_q, \hat{y}_j W_k \rangle / \sqrt{d}\big)$$
$$= Softmax\big(\hat{y}_i (W_q W_k^T) \hat{y}_j^T / \sqrt{d}\big) = Softmax\big(\theta \cdot \langle \hat{y}_i, \hat{y}_j \rangle / \sqrt{d}\big) \tag{18}$$

Furthermore, since LayerNorm preserves the direction of vectors by normalizing only their magnitude, the inner product between normalized outputs satisfies $\langle \hat{y}_i, \hat{y}_j \rangle = \cos(\angle(y_i, y_j))$,

$$\alpha_{ij} = Softmax\big(\theta \cdot \cos(\angle(y_i, y_j)/\sqrt{d}\big) \propto \cos(\angle(y_i, y_j)) \quad (19)$$

Equation 19 indicates that the more aligned the hidden states of two input tokens are (i.e., the smaller the angle between them), the higher their attention score.

Therefore, in conjunction with the layer-wise angle concentration pattern discussed in the main text, we observe that inter-segment angular concentration reflects the model's degree of attention to internal components of the problem, while intra-segment angular concentration captures the level of attention between the question and the system prompt—serving as an indicator of the model's instruction-following capability. $\qquad\square$

*Theoretical Insight3. Presence of sink attention promotes angular concentration among hidden states.*

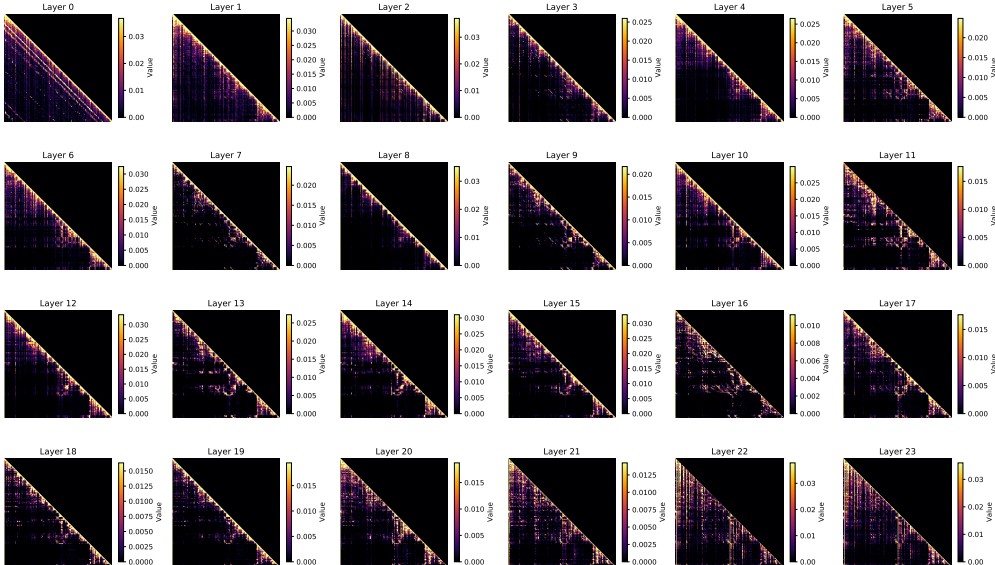

Figure 15: Visualization of attention score at different layers in Qwen2.5-0.5B-Instruct.

*Justification.* To understand why token angles exhibitLayer-wise angular concentration patterns, we start our analysis from the phenomenon of *sink attention*, which similarly shows segment-wise characteristics yet remains insufficiently understood. In LLMs, attention scores exhibit segment-wise tendencies, and, apart from self-attention, attention scores typically peak at the first token of each segment. This phenomenon is termed *sink attention*, as illustrated in Figure 15.

Given an input representation for an attention block in an LLM as $y \in \mathbb{R}^{m \times d}$, where the $i$-th token is denoted as $y_i$, the attention mechanism output is defined as $O = \alpha V$, where $\alpha$ represents the attention scores, and $V = \text{LayerNorm}(y)W_v$ denotes the value vectors. Suppose the attention scores within a segment "sink" to the first token vector $y_i$, then the outputs for the sink token $y_i$ and another token $y_k$ within the same segment can be approximated as:

$$O_i \approx \alpha_{ii}V_i, \quad O_k \approx \alpha_{ik}V_i + \alpha_{kk}V_k,$$

where $\alpha_{ik}$ denotes the attention score between tokens $i$ and $k$. This approximation arises because the sink token's self-attention score significantly surpasses its attention scores to other tokens, while other tokens primarily attend to themselves and the sink token.

Substituting these approximations, the angle between outputs $O_i$ and $O_k$ can be expressed as:

$$\cos(\angle(O_i, O_k)) = \frac{\alpha_{ik}\|V_i\| + \alpha_{kk}\|V_k\| \cos(\angle(V_i, V_k))}{\sqrt{\alpha_{ik}^2\|V_i\|^2 + \alpha_{kk}^2\|V_k\|^2 + 2\alpha_{ik}\alpha_{kk}\|V_i\|\|V_k\| \cos(\angle(y_i, y_k))}}$$

Furthermore, since $W_v$ is approximately an orthonormal matrix (detailed proof in the appendix) and thus preserves angles, we have $W_v W_v^\top \approx \beta I$, where $\beta$ is a constant. Combined with the fact that

LayerNorm scales only magnitudes without altering angles, we get:

$$\cos(\angle(\boldsymbol{V}_i, \boldsymbol{V}_k)) = \cos(\angle(\boldsymbol{y}_i, \boldsymbol{y}_k)).$$

Substituting into the earlier expression, we derive:

$$\cos(\angle(\boldsymbol{O}_i, \boldsymbol{O}_k)) = \frac{\alpha_{ik} + \alpha_{kk} \cos(\angle(\boldsymbol{y}_i, \boldsymbol{y}_k))}{\sqrt{\beta}\sqrt{\alpha_{ik}^2 + \alpha_{kk}^2 + 2\alpha_{ik}\alpha_{kk} \cos(\angle(\boldsymbol{y}_i, \boldsymbol{y}_k))}}$$

Squaring both sides and simplifying, we arrive at the condition:

$$\cos(\angle(\boldsymbol{O}_i, \boldsymbol{O}_k)) > \cos(\angle(\boldsymbol{y}_i, \boldsymbol{y}_k)) \quad \text{if} \quad \beta(\cos(\angle(\boldsymbol{y}_i, \boldsymbol{y}_k)))^2 < 1 \quad \text{and} \quad \beta < 1.$$

Since weight parameters in LLMs are typically constrained to values less than 1, both $\beta < 1$ and $\beta(\cos(\angle(\boldsymbol{y}_i, \boldsymbol{y}_k)))^2 < 1$ generally hold. Thus, Equation (7) demonstrates that sink attention inherently promotes angle concentration within segments. A detailed derivation is provided in the appendix. □

In Figure 15, we present the distribution of attention scores across different layers. In intermediate layers, sink tokens operate primarily within segments to enhance intra-segment angle concentration. In later layers, sink tokens across segments begin to interact, promoting inter-segment concentration. This indicates that, due to the influence of sink tokens, token angles are increasingly concentrated through the forward pass. The final layer's angle concentration is particularly important as it reflects the culmination of this process and directly determines the model's output.

## B.3   Neuron-Based Explanation of Data-wise Angle Concentration Patterns

In Section 2.3 of the main text, we demonstrate that the greater the number of tokens activating the same neuron, the more gradient components that neuron receive. In this section, we further show that the number of commonly activated neurons between tokens has a direct effect on the angle between their output hidden states at the current layer.

*Theoretical Insight 4. Within the FFN block, the extent of overlap in activated neurons between two tokens directly affects the angular relationship between their output hidden states.*

*Justification.* To analyze how the activation layer affects $\cos(\angle(y_i, y_M))$, we first decompose its computation formula. Suppose the activation output of the $i$-th token is $A_i$, and $y_i = A_i W_d$. Based on Observation 1, $W_d$ is a scalar multiple of a unitary self-orthogonal matrix, applying the same rotation to any input while preserving the inner product and angle between any two input vectors. Thus, we can have:

$$\begin{aligned}
\cos(\angle(y_i, y_M)) &= \langle A_i W_{\mathrm{d}}, A_M W_{\mathrm{d}} \rangle / (\|y_i\|\|y_M\|) \\
&= A_i(W_{\mathrm{d}} W_{\mathrm{d}}^T) A_M^T / (\|y_i\|\|y_M\|) \\
&= \eta \langle A_i, A_M \rangle / (\|y_i\|\|y_M\|),
\end{aligned} \tag{20}$$

where $\eta$ is a constant based on Observation 1. Furthermore, since $W_d$ is an orthogonal matrix,

$$\begin{aligned}
\|y_i\|^2 &= \|A_i W_d\|^2 = (A_i W_d)(A_i W_d)^T \\
&= A_i(W_d W_d^T) A_i^T = \eta A_i A_i^T = \eta \|A_i\|^2.
\end{aligned} \tag{21}$$

which means $\|y_i\| = \sqrt{\eta}\|A_i\|$. Substituting this into Eq. 20 we can have

$$\begin{aligned}
\cos(\angle(y_i, y_M)) &= \eta \langle A_i, A_M \rangle / (\eta \|A_i\|\|A_M\|) \\
&= \cos(\angle(A_i, A_M))
\end{aligned} \tag{22}$$

This shows that the orthogonal matrix $W_d$ does not change the angles between the input vectors. Furthermore, based on Observation 2, $cos(\angle(A_i, A_M)) \propto |\Gamma(x_i) \cap \Gamma(x_M)|$, we can get

$$\cos(\angle(y_i, y_M)) \propto |\Gamma(x_i) \cap \Gamma(x_M)|. \tag{23}$$

which is consistent with Insight 2. □

Eq. 23 shows that activation layers adjust token angles by controlling the intersections of their activated neurons. More shared activated neurons lead to smaller angles and greater mutual influence.

# C Visualization Results

In the main text, we presented layer-wise, epoch-wise, and data-wise patterns of angular concentration. In this section, we provide more comprehensive visualizations to further support our conclusions.

## C.1 Layer-wise Angle Concentration Patterns

In Fig. 16, 17 and 18, we present the layer-wise angular concentration patterns across all layers of the Qwen2.5-0.5B-Instruct model for tasks of easy, medium, and high difficulty, respectively. Notably, the model consistently demonstrates first intra-segment angle concentration and subsequently inter-segment angle concentration—regardless of problem difficulty. This consistency suggests that the observed pattern is a generalizable property of the model's internal representation dynamics.

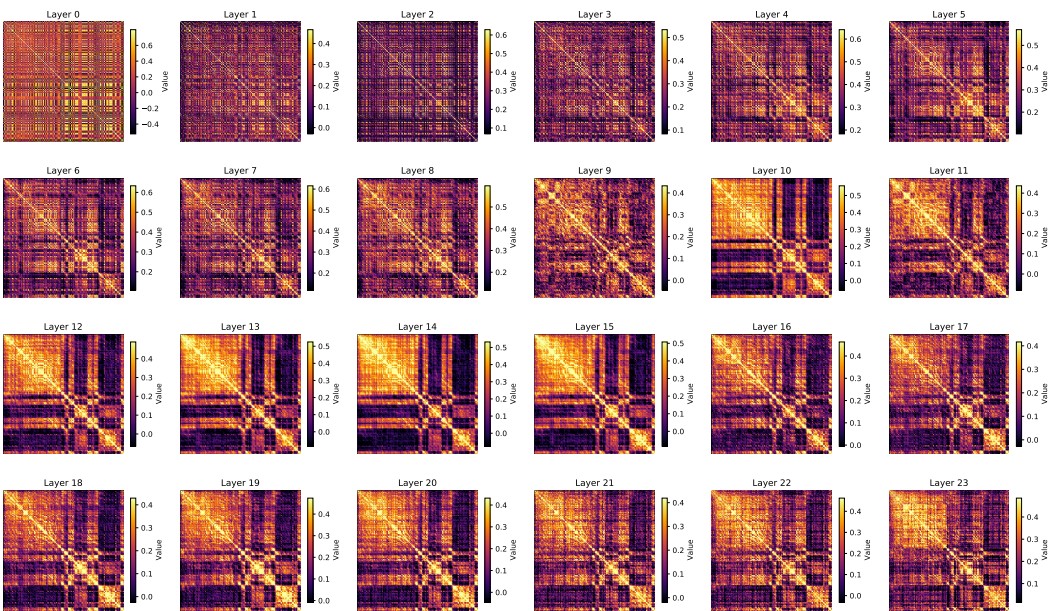

Figure 16: Visualization of Layer-wise Angle Concentration at different layers in Qwen2.5-0.5B-Instruct at easy sample (correct num = 10).

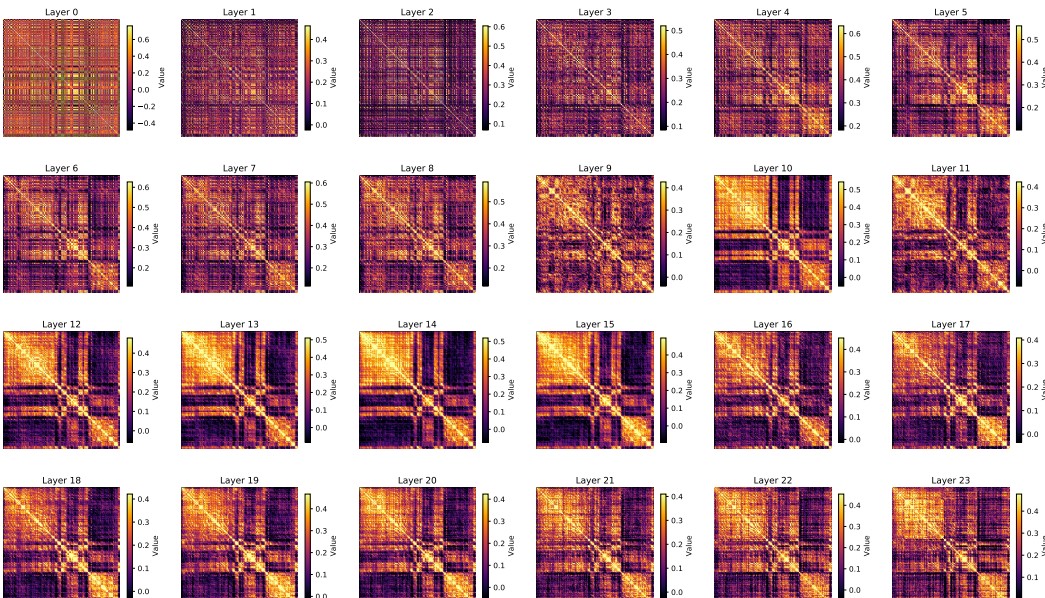

Figure 17: Visualization of Layer-wise Angle Concentration at different layers in Qwen2.5-0.5B-Instruct at medium difficulty sample (correct num = 5).

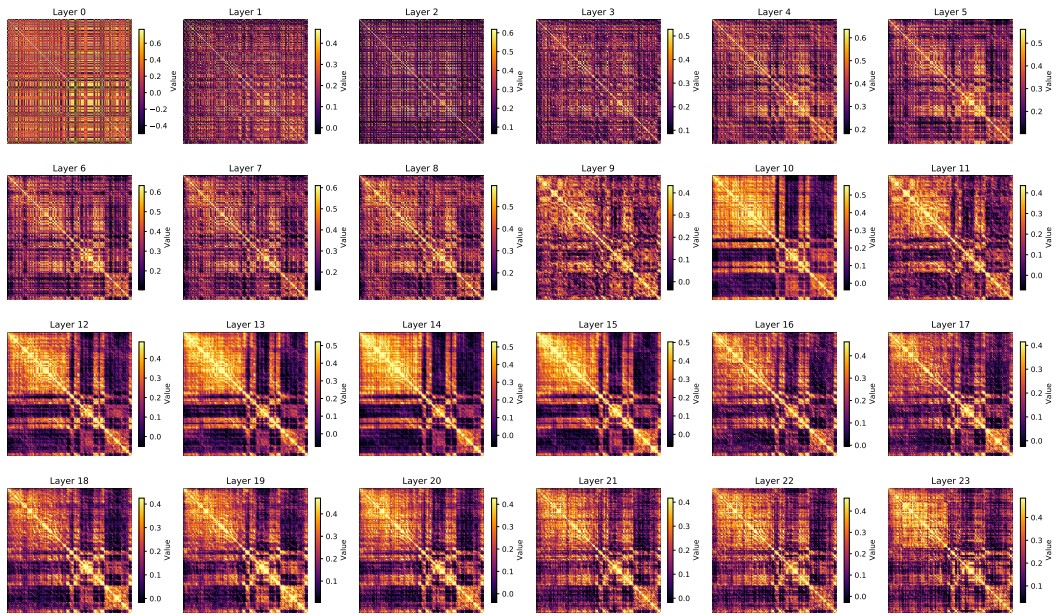

Figure 18: Visualization of Layer-wise Angle Concentration at different layers in Qwen2.5-0.5B-Instruct at difficult sample (correct num = 0).

## C.2 Data-wise Angle Concentration Patterns

Fig. 19 provides a more fine-grained view of the data-wise angle-concentration patterns. Consistent with the conclusions in the main text, it reveals a curriculum-like trend: the model learns samples exhibiting high angular concentration earlier, followed by samples with lower angular concentration.

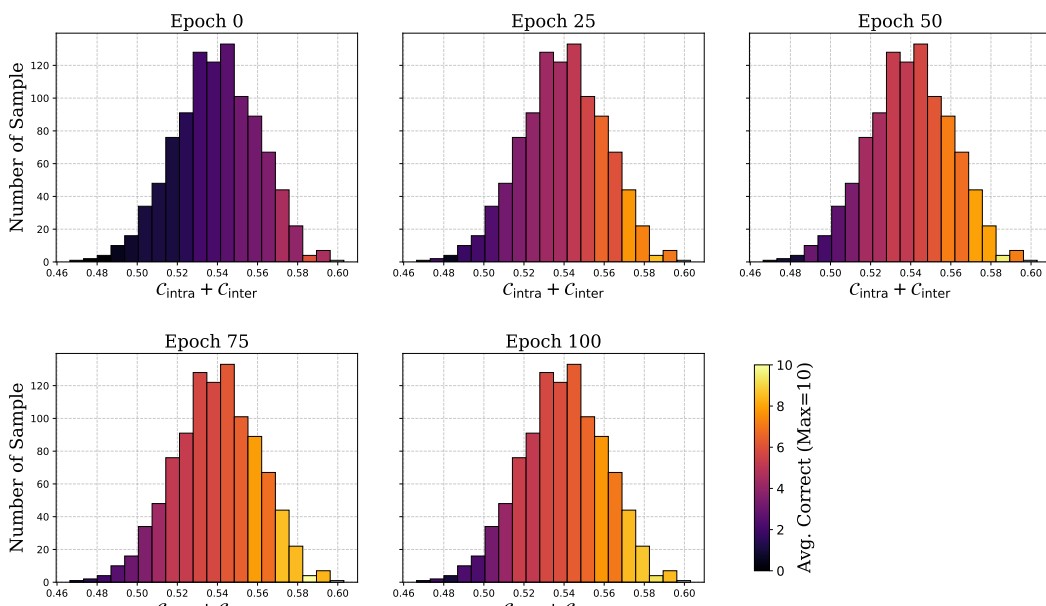

Figure 19: Visualization of Data-wise Angle Concentration at different iterations in Qwen2.5-0.5B-Instruct at difficult sample. The training setting is consistent with Figure 3 in the main text.

# D  GAIN-RL Algorithm

---

**Algorithm 1:** GAIN–RL (GRPO): Gradient-driven Angle-Informed Navigated RL Framework

---

**Input:** Training data $D = \{d_1, d_2, \ldots, d_N\}$; model $M$; training steps $T$;
batch size $n$; accuracy sensitivity $\alpha$; target accuracy $\beta$;
angle sensitivity $\gamma$; sampling variance $\sigma$
**Output:** Trained model $M_T$

**1** **Step 1: Reorder training data by angle signal**
**2** **for** $i \leftarrow 1$ **to** $N$ **do**
**3** $\quad$ Prefill $d_i$ with model $M$
**4** $\quad$ Extract angle signal $\mathcal{C}_M(d_i) = \mathcal{C}^M_{\text{intra}}(d_i) + \mathcal{C}^M_{\text{inter}}(d_i)$
**5** Sort $D$ by $\mathcal{C}_M(\cdot)$ in descending order to obtain $D_s$

**6** **Step 2: Train with dynamic probabilistic sampling**
**7** Initialize Gaussian distribution $P_0 \sim \mathcal{N}(0, \sigma^2)$
**8** **for** $t \leftarrow 1$ **to** $T$ **do**
**9** $\quad$ Sample batch $d^{(t)} \sim \text{Sample}(D_s; P_t, n)$
**10** $\quad$ **for** $i \leftarrow 1$ **to** $n$ **do**
**11** $\quad\quad$ Let model $M_t$ answer $d_i^{(t)}$
**12** $\quad\quad$ Record $\mathcal{C}_{M_t}(d_i^{(t)})$ and accuracy $\text{Acc}_{M_t}(d_i^{(t)})$
**13** $\quad$ Update $M_t$ with accuracy-based loss
**14** $\quad$ Compute mean accuracy $\text{Acc}^{(t)} = \frac{1}{n} \sum_{i=1}^{n} \text{Acc}_{M_t}(d_i^{(t)})$
**15** $\quad$ Compute mean angle $\mathcal{C}^{(t)} = \frac{1}{n} \sum_{i=1}^{n} \mathcal{C}_{M_t}(d_i^{(t)})$
**16** $\quad$ Update sampling-mean

$$\mu_{t+1} = \mu_t + \frac{n}{2} \tanh\big(\alpha(\text{Acc}^{(t)} - \beta)\big) + \frac{n}{2} \tanh\big(\gamma\, \mathcal{C}^{(t)}\big)$$

---

# E  Experimental Setup

In this section, we describe our experimental setup in detail, covering the models, datasets, and hyperparameters used.

## E.1  Model and Dataset

To comprehensively evaluate the effectiveness of GAIN-RL, we conduct experiments across multiple models and datasets. Specifically, we select models varying in size, including Qwen2.5-0.5b-Instruct, Qwen2.5-Math-1.5B-Instruct, Qwen2.5-Math-7B-Instruct, Qwen2.5-Coder-3B-Instruct, and LLaMA3.2-3B-Instruct. We primarily focus on two tasks: Math and Code. To evaluate the training efficiency of GAIN-RL (Section 4.1 in the main text), we use the DeepScaleR [24] dataset for mathematical task training and DeepCoder [25] for coding tasks training, each integrating problems from diverse sources and covering a wide range of difficulty levels. For mathematical evaluations, we employed six benchmark datasets of varying difficulty: GSM8K [3], MATH [4], AMC 23 [18], AIME 24 [19], OlympiadBench [20], and Minerva Math [48]. For coding evaluations, we utilized three standard benchmark datasets: LivecodeBench (8/1/24–2/1/25) [6], Codeforces [22], and Humaneval+ [23]. For other experiments (Section 4.2-Section 4.5), we train model on the training dataset of single tasks including GSM8K, MATH and AMC 23 to facilitate more convenient comparisons.

## E.2  Training Configuration

We trained the models using the GRPO algorithm. The training was performed on a single node equipped with 8 A100 GPUs. Each model was trained for about 200 steps using the veRL library.

To evaluate the training efficiency on GRPO-RL, the main training configuration for Qwen2.5-Math-7B-Instruct is shown below. For Qwen2.5-Math-1.5B-Instruct and LLaMA3.2-3B-Instruct, we set `max_response_length=3000` to accommodate its shorter context window of 4096 tokens, while keeping all other parameters unchanged. For Qwen2.5-0.5B-Instruct and single task training, we set `max_prompt_length=max_response_length=512`, while keeping other parameters unchanged. For Qwen/Qwen2.5-Coder-3B-Instruct, we set `max_prompt_length=2048`, `max_response_length=16384,train_batch_size=512` and `ppo_mini_batch_size=64` due to its higher single sample memory usage.

```
python3 -m verl.trainer.main_ppo \
    algorithm.adv_estimator=grpo \
    data.train_files="$train_files" \
    data.val_files="$test_files" \
    data.train_batch_size=1024 \
    data.max_prompt_length=1024 \
    data.max_response_length=8192 \
    actor_rollout_ref.model.path=Qwen/Qwen2.5-Math-7B-Instruct \
    actor_rollout_ref.actor.optim.lr=1e-6 \
    actor_rollout_ref.model.use_remove_padding=True \
    actor_rollout_ref.actor.ppo_mini_batch_size=256 \
    actor_rollout_ref.actor.ppo_micro_batch_size_per_gpu=16 \
    actor_rollout_ref.actor.use_dynamic_bsz=True \
    actor_rollout_ref.actor.ppo_max_token_len_per_gpu=8000 \
    actor_rollout_ref.actor.use_kl_loss=True \
    actor_rollout_ref.actor.kl_loss_coef=0.001 \
    actor_rollout_ref.actor.kl_loss_type=low_var_kl \
    actor_rollout_ref.actor.entropy_coeff=0 \
    actor_rollout_ref.model.enable_gradient_checkpointing=True \
    actor_rollout_ref.actor.fsdp_config.param_offload=False \
    actor_rollout_ref.actor.fsdp_config.optimizer_offload=False \
    actor_rollout_ref.rollout.tensor_model_parallel_size=1 \
    actor_rollout_ref.rollout.log_prob_micro_batch_size_per_gpu=16 \
    actor_rollout_ref.rollout.name=vllm \
    actor_rollout_ref.rollout.gpu_memory_utilization=0.6 \
    actor_rollout_ref.rollout.n=8 \
    actor_rollout_ref.ref.log_prob_micro_batch_size_per_gpu=16 \
    actor_rollout_ref.ref.fsdp_config.param_offload=True \
    algorithm.use_kl_in_reward=False \
    trainer.critic_warmup=0 \
    trainer.logger=['console','wandb'] \
    trainer.project_name='verl_grpo_example_gsm8k_math' \
    trainer.experiment_name='qwen2_MATH_7b_Instruct_function_rm' \
    trainer.n_gpus_per_node=8 \
    trainer.nnodes=1 \
    trainer.save_freq=20 \
    trainer.test_freq=5 \
    trainer.total_epochs=200 $@
```

# F   Additional Experimental Results

In this section, we present additional experiments to further validate the effectiveness of GAIN-RL.

## F.1   Performance of Weighted Signals

In the main text, we employed an unweighted angular signal of the form: $\mathcal{C}_M(d_i) = \mathcal{C}_{\text{intra}}^M(d_i) + \mathcal{C}_{\text{inter}}^M(d_i)$. Here, we explore a weighted variant of the signal: $\mathcal{C}_M(d_i) = \mathcal{C}_{\text{intra}}^M(d_i) + c \cdot \mathcal{C}_{\text{inter}}^M(d_i)$, where $c$ is a constant weight.

As shown in Tab. 4, we investigate the final performance of the Qwen2.5-0.5B-Instruct model after 200 training iterations on the Math dataset, using different values of $c$ in the weighted signal. The results show that setting $c = 4.0$ yields the best performance for this model, suggesting that carefully designed angle-based signals can further improve training effectiveness.

However, selecting the optimal weighting coefficient $c$ requires extensive empirical tuning. To ensure scalability and practical applicability, we adopt the unweighted version of the signal in this work and leave signal optimization for future research. Our goal is to demonstrate that *even without finely tuned weighting*, GAIN-RL is still capable of accelerating both training and data efficiency—highlighting the strong potential of model-signal-based RLHF strategies.

Table 4: Model performance at iteration 200 under different weighting coefficients $c$.

| $c$ | 0.25 | 0.5 | 1.0 | 2.0 | 4.0 | 8.0 |
|---|---|---|---|---|---|---|
| Accuracy | 36.20 | 37.20 | 37.40 | 38.00 | 38.40 | 38.20 |

## F.2 Model Performance on Single Task

Fig. 9 in the main text illustrates the training dynamics of Qwen-2.5-0.5B-Instruct on three single-task datasets. For a more detailed comparison, Tab. 5 reports the final performance at iteration 200 and the corresponding training speedup across different training sets and methods. Notably, on the GSM8K dataset, GAIN-RL outperforms the original GRPO by 4.72% in final accuracy and achieves a $3.3\times$ improvement in training speed.

These results demonstrate that GAIN-RL can effectively distinguish between samples of varying learnability even in the single-task setting, highlighting its efficiency and general applicability.

Table 5: Fine-tuning performance on single tasks. Models are trained on the training sets and evaluated on their validation sets. ADARFT is excluded on GSM8K due to missing difficulty coefficients.

| | Prepare | | GSM8K | | | Math | | | AMC | | |
|---|---|---|---|---|---|---|---|---|---|---|---|
| | Metric | Time | ACC@ 200Iter | Iter@ 200Acc | Speed Up | ACC@ 200Iter | Iter@ 200Acc | Speed Up | ACC@ 200Iter | Iter@ 200Acc | Speed Up |
| GRPO | - | - | 48.43 | 200 | 1× | 34.80 | 200 | 1× | 9.64 | 200 | 1× |
| ADARFT(GRPO) | Difficulty | > 1 day | - | - | - | 35.80 | 150 | 1.33× | 9.64 | 160 | 1.25× |
| **GAIN-RL(GRPO)** | **Angle** | **< 10 min** | **53.15** | **60** | **3.33×** | **37.40** | **80** | **2.50×** | **12.04** | **100** | **2.00×** |

# G Discussion and Future Work

In Section 3, we demonstrate that angles between token hidden states fundamentally mirror and influence both the information propagation during inference and the learning dynamics throughout model training. The proposed angle-based signals can, in fact, be generalized beyond RFT to enhance model-centric effectiveness in various other domains. For instance, during pre-training, monitoring angle signals could enable real-time evaluation of a model's learning capacity across different domains, thus allowing adjustments to training data to improve stability and final performance. Furthermore, during inference, tracking changes in angle concentration between layers could provide insights into the model's comprehension of inputs and indicate whether additional test-time adjustments are necessary to boost output accuracy. In future work, we plan to further investigate how this signal can be leveraged across multiple domains to achieve comprehensive, model-centric optimizations.

