# OpenReview forum: "Angles Don’t Lie: Unlocking Training‑Efficient RL Through the Model’s Own Signals"
_NeurIPS.cc/2025/Conference — NeurIPS 2025 spotlight_

### Official Review · Reviewer_LV4y · 2025-07-01

**Clarity:** 4
**Significance:** 3
**Originality:** 3
**Rating:** 5
**Confidence:** 3

**Summary:**

Authors present an interesting observation on LLM training dynamics: the Qwen2.5-0.5B-Instruct model prioritizes learning samples with high angle score ($C_{intra}+C_{inter}$) and progressively learns lower-score samples. The angle score has both theoretical and empirical connection to the attention score between question tokens and between question tokens and instruction/example tokens.

With this observation in hand, the paper proposes a novel framework that skews the sample distribution to favor those with high angle score, updating the distribution to lower scores as training progresses. They were able to achieve significant improvements in sample efficiency with virtually no computation overhead made in the dataset preprocessing (sorting) phase.

**Questions:**

1. As stated in the weaknesses, "High AngConc-biased Sampling" seems to perform better than GAIN-RL and is much simpler (e.g., less hyperparameters). I cannot help but ask: why not just use that?

2. In eq8, how is $\sigma_t$ tuned?

3. Looking at eq10, it seems like $\mu_t$ starts from 0 then increases indefinitely. Are there guarantees that $\mu_t$ doesn't exceed $n$, or is it explicitly forced to be under $n$?

4. As $\mu_t$ increases, would samples with low index (high angle scores) eventually have zero probability? If so, wouldn't it cause issues for large datasets such as catastrophic forgetting?

**Ethical Concerns:**

["NO or VERY MINOR ethics concerns only"]

**Final Justification:**

The main problem was that the writing was a bit confusing and difficult to read. I think this has been sufficiently addressed.

**Limitations:**

Authors do discuss future work in Appendix.G but not specifically their limitations.

**Paper Formatting Concerns:**

Vspace seems to be abused, which hopefully gets revised shortly.

**Quality:**

3

**Strengths And Weaknesses:**

Strengths
- The main observation/motivation is interesting, and is supported by theoretical and empirical evidences.
- The experimental setup is clear and straighforward, and their results support their aforementioned observation across diverse LLM models and sizes.

Weaknesses
- I would have been nice to see experiment on much larger models (at least on few menchmarks), although I believe 7B meets the standard.
- Figure8 seems to reveal that angle-biased sampling is a more simple yet superior method to GAIN-RL, which I think needs more explanation than what's said on line 278-280.

---

> ### Author Rebuttal · Authors · 2025-07-31
>
> We thank the reviewer for the positive and encouraging feedback. We’re glad that the motivation and supporting evidence were found compelling, and that the experimental setup was seen as clear and effective. We hope the following responses further address any remaining questions.
>
>
>
> > ***Weakness1. I would have been nice to see experiment on much larger models (at least on few menchmarks), although I believe 7B meets the standard.***
>
> **Responce**:  We thank the reviewer for the helpful suggestion. We fully agree that experiments on larger models would further highlight the effectiveness and scalability of our approach. In fact, we have already begun running experiments on larger models and plan to include the results in the discussion section or in an updated version of the paper.
>
>
>
> > ***Weakness2. Figure8 seems to reveal that angle-biased sampling is a more simple yet superior method to GAIN-RL, which I think needs more explanation than what's said on line 278-280.***
>
> **Responce**: We sincerely thank the reviewer for this thoughtful and insightful comment. As the reviewer rightly pointed out, Figure 8 deserves a more nuanced explanation. We fully agree and are happy to provide a deeper analysis of this promising result.
>
> Figure 8 illustrates the effect of our proposed High Angular Concentration-biased Sampling method, which selectively excludes samples with low angular concentration. The results suggest that **removing such samples can significantly improve training efficiency while preserving gradient effectiveness and stability.** This observation is supported by several layers of evidence:
>
> 1. **Theoretical justification via Equation (4):** Low angular concentration samples reduce the mean and increase the variance of training gradients. As shown in Equation (4), angular concentration among hidden states directly influences the Frobenius norm of the gradient through the sum of cos θ terms. A higher angular concentration yields stronger, more consistent gradient updates, accelerating the learning process and improving efficiency.
> 2. **Empirical support from the neuron activation patterns (Figure 6):** As shown in Figure 6, high-angular-concentration samples tend to activate a focused cluster of core neurons, enabling effective gradient accumulation and rapid formation of high-performing neuron clusters. In contrast, low-concentration samples activate more scattered neurons, leading to dispersed gradient directions. Although more samples tend to activate core neurons as training progresses, a subset of low-concentration samples continues to produce noisy and inconsistent activations, thereby reducing overall training stability and efficiency.
> 3. **Alignment with prior research findings:** Several prior works [1,2,3] have shown that not all training samples contribute positively to learning, and removing "negative gain" samples can lead to faster and more stable optimization. Our method provides a practical and computationally efficient way to filter out such low-value data points based on angular concentration.
>
> We will expand our discussion of this result in the revised version and plan to explore this direction further in future work.
>
> Once again, we sincerely appreciate the reviewer’s constructive feedback, which has helped us improve the depth and completeness of our paper.
>
>
>
> > ***Question 1. As stated in the weaknesses, "High AngConc-biased Sampling" seems to perform better than GAIN-RL and is much simpler (e.g., less hyperparameters). I cannot help but ask: why not just use that?***
>
> **Responce**:  We sincerely thank the reviewer for raising this insightful question. We appreciate the recognition of the effectiveness of high angular concentration-based sampling and are happy to clarify why we adopt GAIN-RL over a simpler hard-filtering approach.
>
> 1. **Angular concentration is dynamic during training:** As model capacity improves, samples with initially low angular concentration often evolve into high-concentration ones (see Figure 4 in the main text). These examples, though have low angular concentration and weak early on, may contain long-tail semantics or rare states that become high angular concentration and valuable later. Therefore, filtering based on static pre-training concentration may prematurely discard useful data.
> 2. **Static thresholds lack generalizability; GAIN-RL offers a dynamic solution:** While Figure 8 shows promising results using a fixed top 50% threshold, this cutoff is empirical and dataset-specific. Angular concentration distributions vary widely across datasets, and different settings may require different thresholds. In contrast, GAIN-RL introduces a dynamic, model-aware sampling strategy that avoids trial-and-error tuning and generalizes better across tasks and architectures.
> 3. **Hard data removal can be harmful:** In large-scale RL + LLM training, completely discarding data risks overfitting to high-concentration clusters or triggering training collapse. Prior studies [1,2,3] also suggest that importance-weighted sampling is more robust and generalizable than hard filtering. Even when de-emphasizing low-concentration samples, it is often better to retain them with down-weighted importance than to remove them entirely.
>
> While we agree that angular concentration is a powerful signal for data selection, the current static filtering setup (i.e., dropping half the data) requires further validation to assess its generalization. We plan to include these results in the discussion section or an updated version of the paper. In contrast, GAIN-RL has already demonstrated strong generalization across datasets and model sizes, making it a more practical and robust choice.
>
> Once again, we sincerely thank the reviewer for the insightful comment, which helped us clarify and strengthen our position.
>
>
>
> > ***Question 2. In eq8, how is $\sigma_t$ tuned?***
>
> **Responce**:   We sincerely thank the reviewer for the careful reading and for raising this important question. We apologize for the omission of the explanation regarding the choice of $\sigma_t$ in the paper, and we are happy to clarify it here.
>
> In our implementation, we set $\sigma_t$ as a fixed value: $\sigma_t = N/3$ , where $N$ is the total number of training samples. This choice is motivated by both theoretical considerations and empirical ablation studies:
>
> **Theoretical motivation:** To avoid catastrophic forgetting during training, we aim to ensure that all samples maintain a non-zero probability of being selected at any point. For a Gaussian distribution, 99.73% of the mass lies within $[\mu - 3 \sigma, \mu + 3 \sigma]$. By setting $\sigma_t = N/3$, even in the most extreme case (when $\mu_t = N$), the earliest 10% of samples—those with the highest angular concentration—still retain approximately 7% sampling probability. This helps preserve important early-stage knowledge throughout training.
>
> **Empirical validation:** We conducted ablation experiments on the MATH dataset using Qwen2.5-0.5B-Instruct, evaluating model performance after 200 epochs under different values of σ. The results show that: A smaller $\sigma_t$ (e.g., $N / 5$) leads to sharp forgetting, with validation accuracy rising initially but then degrading due to over-focusing on a narrow sample range. A larger $\sigma_t$ (e.g., $N / 2$) slows down training by diluting the prioritization of high-gradient samples. $\sigma_t = N / 3$ strikes a good balance, enabling both efficient training and knowledge retention.
>
> | Sigma Value | N/2   | N/3   | N/4   | N/5   |
> | ----------- | ----- | ----- | ----- | ----- |
> | Accuracy    | 21.2% | 22.6% | 22.5% | 19.3% |
>
> We will include a detailed explanation and ablation results regarding the choice of σ in the revised version. We sincerely appreciate the reviewer’s helpful comment, which has allowed us to clarify this important design decision.
>
>
>
>
>
> > ***Question 3. Looking at eq10, it seems like $\mu_t$ starts from 0 then increases indefinitely. Are there guarantees that $\mu_t$ doesn't exceed , or is it explicitly forced to be under n?***
>
> **Responce**:  We sincerely thank the reviewer for the helpful question. In our implementation, we explicitly enforce a truncation on $\mu_t$ such that $\mu_t = min(\mu_t, N)$, where $N$ is the total number of training samples. Once $\mu_t$ reaches $N$, the current training round is terminated at the end of that epoch, and a new round is restarted with $\mu_t$ reset to 0. This design ensures that sampling remains valid and the training process remains effective throughout.
>
> We will clarify this implementation detail in the revised version. We appreciate the reviewer’s question, which helps improve the completeness of our explanation.
>
>
>
>
>
> > ***Question 4. As $\mu_t$ increases, would samples with low index (high angle scores) eventually have zero probability? If so, wouldn't it cause issues for large datasets such as catastrophic forgetting?***
>
> **Responce**:  We sincerely thank the reviewer for raising this important concern. As mentioned in our response to Question 2, our $\sigma_t$ value is set relative to the total number of training samples ($\sigma_t = N/3$), which ensures that all samples retain a non-zero probability of being selected. This design helps effectively prevent catastrophic forgetting, even as $\mu_t$ increases during training [4].
>
>
>
> **References:**
>
> *[1] Schaul T, Quan J, Antonoglou I, et al. Prioritized experience replay[J]. arXiv preprint arXiv:1511.05952, 2015.*
>
> *[2] Pleiss L S, Sutter T, Schiffer M. Reliability-Adjusted Prioritized Experience Replay[J]. arXiv preprint arXiv:2506.18482, 2025.*
>
> *[3] Isele D, Cosgun A. Selective experience replay for lifelong learning[C]//Proceedings of the AAAI conference on artificial intelligence. 2018, 32(1).*
>
> *[4] Lopez-Paz, David, and Marc'Aurelio Ranzato. "Gradient episodic memory for continual learning." *Advances in neural information processing systems* 30 (2017).*

---

### Official Review · Reviewer_ecE4 · 2025-07-01

**Clarity:** 3
**Significance:** 3
**Originality:** 3
**Rating:** 5
**Confidence:** 4

**Summary:**

The paper proposes a novel RFT framework for large language models, called GAIN-RL, which enhances training efficiency by leveraging a model-intrinsic signal called *angle concentration*. This metric quantifies the model’s potential to learn from a given prompt. It is both model-based and computationally efficient, requiring no reliance on heuristic measures of question difficulty. Empirical results demonstrate that angle concentration is a measurable and effective signal for guiding reinforcement learning.

**Questions:**

1. Why is Gaussian probability applied? Is it because Gaussian is very light tailed such that only certain $d_i^s$s will be chosen with high probability? More explanation would be helpful.
2. What’s $\sigma_t$ in (8)?
3. Is this a typo?: the left equation in Eq 8 only includes $i$ instead of $d_i^s$.

**Ethical Concerns:**

["NO or VERY MINOR ethics concerns only"]

**Final Justification:**

The authors fully addressed my concerns about paper clarity. This is a good paper.

**Limitations:**

The observations regarding angle concentration are primarily based on the 0.5B model. It would strengthen the claims and improve the generalizability of the findings if similar analyses were also conducted on larger models.

**Quality:**

3

**Strengths And Weaknesses:**

The paper addresses an important and interesting question, i.e., how to improve sample efficiency and accelerate training in reinforcement learning for large language models. It is widely recognized that selecting appropriate prompts at different stages of training plays a critical role in enhancing both the speed and robustness of RL. The authors introduce a novel, model-based metric that avoids expensive data preprocessing and can effectively guide a new training strategy, GAIN-RL. This strategy is shown to outperform vanilla RL methods based on uniform sampling. Sufficient results confirm the effectiveness of GAIN-RL.

While the paper is well-written and clearly structured, I still have some suggestions and questions about the motivation part, i.e. section 2.
1. The role of the gradient norm could be clarified. While I understand that small gradients might suggest proximity to a solution, it is less obvious why large gradients necessarily imply the opposite. As a first-time reader, I found it unclear why the Frobenius norm of the gradient should be the primary focus.
2. Equation (4) does not clearly isolate the role of the angle in $l_2$ space. It is not fully explained why the norm of the hidden states can be disregarded. Are these norms assumed to be fixed? If not, then both the angle and the magnitudes contribute to the Frobenius norm. Moreover, it seems to me that the angle in equation 4 comes from the Frobenius norm. If another norm were used, then the angle in the $l_2$ norm space doesn’t always show explicitly in the equation.

I suggest the authors revise this section to provide more intuitive explanations and to make the motivation more accessible to a broader audience.

---

> ### Author Rebuttal · Authors · 2025-07-31
>
> We sincerely thank the reviewer for the kind and encouraging feedback. We are honored that the reviewer considers the problem we address to be important and recognizes the effectiveness of our proposed GAIN-RL strategy. We greatly appreciate the positive evaluation of our contributions and the time and effort dedicated to reviewing our work. We hope the following responses provide further clarification and address any remaining concerns.
>
>
>
> > ***Weakness1. The role of the gradient norm could be clarified.***
>
> **Responce**:  We sincerely thank the reviewer for the thoughtful and constructive feedback. We appreciate the opportunity to further clarify the motivation behind focusing on the gradient Frobenius norm in our work.
>
> **1. Why gradient norm reflects learnability:**
>
> The gradient norm is a fundamental indicator of how far a model is from satisfying the first-order optimality condition. Specifically, if $\nabla f= 0$, then $x$ is a critical point. Conversely, if $\|\|\nabla f\|\|> 0$, it directly quantifies the gap from optimality. In smooth and strongly convex settings, the gradient norm also provides bounds on the distance to the optimum [1,2]. Hence, a large gradient norm indicates that the model still has substantial room to learn, rather than proximity to convergence.
>
> This perspective is widely adopted in prior work across various learning paradigms. For instance, in multi-task learning, GradNorm [3] adjusts task weights based on gradient magnitudes; in training efficiency, Selective Backprop [4] prioritizes samples with large gradients; and in reinforcement learning, gradient norms serve as reward signals for curriculum generation [5]. These works demonstrate the empirical utility of gradient norms as a measure of learnability.
>
> **2. Why we use the Frobenius norm:**
>
> We adopt the Frobenius norm because it is theoretically principled and practically efficient. It treats the gradient tensor as a Euclidean vector and is invariant to orthogonal transformations—an important property for deep networks with parameter symmetries. It also aligns naturally with the geometry of first-order optimizers like SGD and Adam. Unlike spectral norms, it has linear-time complexity and avoids the non-differentiability of L1 norms.
>
> In practice, the Frobenius norm is widely used as the default global gradient clipping metric in frameworks like PyTorch and TensorFlow. It is also used in estimating the gradient noise scale for adaptive batch sizing in large-scale transformer training. These conventions further validate its role as a reliable, well-established metric for capturing how much a model has yet to learn.
>
> We will improve the explanation in the revised version with clearer intuition and motivation to make it more accessible to first-time readers.
>
>
>
> > ***Weakness2. Equation (4) does not clearly isolate the role of the angle in l2 space.***
>
> **Response**: We thank the reviewer for the insightful question about Equation (4). Below we clarify why the angle term is emphasized and why hidden state norms can be treated as approximately constant.
>
> **1. Why the norms $\|\|x_i\|\|$ and $\|\|x_j\|\|$ can be disregarded in Equation (4):**
>
> In Transformer models like Qwen and LLaMA, hidden vector norms are stabilized by LayerNorm, which normalizes vectors and applies learnable scaling ($\gamma$) and bias ($\beta$). The resulting norm, $|y - \beta|_2 = \sqrt{\sum_i \gamma_i^2}$, is nearly constant across samples and evolves slowly during training. With typical hidden sizes ($d \approx 10^3$), the relative variation in norms is under 3%, allowing us to approximate them as constant scaling factors when analyzing gradient norms. This simplification is also supported by prior work [6].
>
> **2. Why we analyze angles in ℓ₂ space (Frobenius norm):**
>
> Angles are meaningful only in inner-product spaces, which the Frobenius norm supports. Non-Euclidean norms like ℓ₁ do not preserve inner products and thus lack a coherent notion of angle. Moreover, analyzing angles under the ℓ₂ norm is both standard and appropriate. As discussed in our response to Weakness 1, the Frobenius norm is widely used in practice, aligns with standard optimization assumptions, and corresponds directly to generalization bounds tied to ℓ₂ regularization. Thus, focusing on angle dynamics within the ℓ₂ geometry provides both interpretability and practical relevance.
>
> We will clarify these points in the revised version and we thank the reviewer for the insightful feedback.
>
>
>
> > ***Question1. Why is Gaussian probability applied?***
>
> **Response**: We sincerely thank the reviewer for the thoughtful question. Our choice of Gaussian is motivated by two key considerations:
>
> 1. **Preventing data forgetting by maintaining non-zero sampling probability**:
>     As the reviewer suggested, Gaussian sampling—with its light tails—ensures that all data points retain a non-zero probability of being sampled. This is crucial for preventing knowledge forgetting during training. In our implementation, we set the standard deviation $\sigma = N/3$ (where $N$ is the total number of training samples). Even in the extreme case where the sampling center μₜ reaches the rightmost end of the data (i.e., $\mu_t = N$), the earliest 10% of the data (with the highest angular concentration) still retains a 7% probability of being sampled. This ensures that previously seen samples remain accessible throughout training.
> 2. **Smooth and controllable transition from high- to low-angle data**:
>    Gaussian sampling enables a dynamic, symmetric distribution centered at $\mu_t$, assigning the highest probability to samples around $d_{\mu_t}$ and gradually decreasing for samples further away. This property allows us to focus the training data distribution around a shifting center, ensuring that at each time step, the model is exposed to a narrow band of samples with relatively similar angular concentration. As $\mu_t$ increases over time, the sampling smoothly transitions from high-concentration to low-concentration samples, which aligns with our intended training curriculum.
>
> Our experiments show that Gaussian sampling consistently accelerates training across models of various sizes, demonstrating both robustness and generalizability. We will further elaborate it in the revised version.
>
>
>
> > ***Question2.  What is $\sigma_t$ in eqa. 8?***
>
> **Response**:  We thank the reviewer for the careful reading and for pointing out this oversight. We apologize for the confusion caused by the missing explanation of $\sigma_t$.
>
> In Equation (8), the sampling probability in GAIN-RL follows a Gaussian distribution with $\mu_t$ and $\sigma_t$ as the mean and standard deviation. While $\mu_t$ increases over time as defined in Equation (10), guiding the sampling center, we fix $\sigma_t = N/3$, where N is the total number of training samples.
>
> This choice ensures that even when $\mu_t = N$, early samples (top 10% angular concentration) still have a notable sampling probability (~7%), helping mitigate knowledge forgetting. We also performed ablation studies on different σ values (see response to Reviewer LV4y, Q2). We will clarify this in the revised version. Thank you again for your helpful feedback.
>
>
>
> > ***Question3.  Is this a typo?: the left equation in Eq 8 only includes $i$ instead of $d_i^s$.***
>
> **Response**: We sincerely thank the reviewer for the careful reading and for raising this question. This is not a typo, and we are happy to provide a more detailed explanation of Equation (8).
>
> GAIN-RL training involves three stages: data reordering, probabilistic sampling, and probability updating. In the reordering stage, all training samples are sorted based on their angular concentration. That is, in the reordered dataset $D_s =d_1^s, d_1^s,..., d_n^s $, a smaller index $i$ indicates higher angular concentration, while a larger $i$ corresponds to lower concentration.
>
> In Equation (8), the left-hand side uses $i$ to denote the index of a sample in the reordered set. This formulation expresses the sampling probability at time step $t$ based on the sample’s position in the ordering. The sample with index $i=\mu_t$ has the highest sampling probability at time $t$, and probabilities decrease symmetrically for samples with indices farther from $\mu_t$. As  $\mu_t$ increases over training (as defined in Equation (10)), the sampling center shifts gradually from high- to low-concentration samples, enabling a smooth curriculum from easy to harder data.
>
> We appreciate the reviewer’s question and will revise the paper to make this interpretation of the index-based sampling clearer in the updated version.
>
>
> > ***Limitation.  Observations on angle concentration should also be conducted on larger models to improve the generalizability of the findings.***
>
> **Response**:  Thank you for the valuable suggestion. We agree that analyzing angular concentration on larger models would enhance the generalizability of our findings. For efficiency, we only presented visualizations on the 0.5B model, but experiments show GAIN-RL consistently accelerates training across models up to 7B parameters, supporting the broader applicability of our insights. We will add visualizations for larger models in the revised version to further support our claims. We appreciate the reviewer’s helpful feedback.
>
>
>
>
>
>
>
> **References:**
>
> [1] Boyd, Stephen P., and Lieven Vandenberghe. *Convex optimization*. Cambridge university press, 2004.
>
> [2] Nesterov, Yurii. *Lectures on convex optimization*. Vol. 137. Berlin: Springer International Publishing, 2018.
>
> [3] Chen, Zhao, et al. "Gradnorm: Gradient normalization for adaptive loss balancing in deep multitask networks.".
>
> [4] Jiang, Angela H., et al. "Accelerating deep learning by focusing on the biggest losers.".
>
> [5] Campbell, Ryan, and Junsang Yoon. "Automatic curriculum learning with gradient reward signals.".
>
> [6]Xiong, Ruibin, et al. "On layer normalization in the transformer architecture.".

---

### Official Review · Reviewer_ERbf · 2025-07-07

**Clarity:** 3
**Significance:** 4
**Originality:** 4
**Rating:** 5
**Confidence:** 3

**Summary:**

This paper introduces GAIN-RL, a novel Reinforcement Learning Fine-tuning (RFT) framework designed to address the sample inefficiency and high computational costs in training Large Language Models. The core insight is the identification of "angle concentration" – the cosine similarity between token hidden states – as a model-inherent signal that effectively reflects an LLM's learning capacity. GAIN-RL leverages this signal by reordering training data based on initial angle concentration and dynamically sampling data during training using a Gaussian distribution, with the mean updated by real-time accuracy and angle signals. Experimental results demonstrate that GAIN-RL significantly accelerates training and achieves better performance with less data compared to vanilla GRPO.

**Questions:**

see in previous comment

**Ethical Concerns:**

["NO or VERY MINOR ethics concerns only"]

**Limitations:**

yes

**Quality:**

3

**Strengths And Weaknesses:**

Strengths
1.	The paper addresses a critical problem of sample inefficiency and high computational costs in LLM RFT, proposing a novel idea of using the model's inherent "angle concentration" signal for data manipulation.

2.	The paper provides strong support for angle concentration as a learning signal through both theoretical reformulation of gradients and empirical observations of various angle concentration patterns.

3.	Experimental results show significant performance and efficiency gains, with GAIN-RL (GRPO) achieving over 2.5x training acceleration and superior performance with half the data.

4.	The framework demonstrates good generalizability, presented as a plug-and-play solution compatible with different models and RL algorithms, with minimal preprocessing cost.

5.	Clear ablation studies effectively demonstrate the contribution of each component, and small-batch scalability tests further validate the method's robustness.
Weaknesses
1.	The claim of "no additional cost" for computing accuracy and angle concentration for probability updates requires more detailed justification across various training setups to ensure consistent fairness.

2.	The paper's strong results with "High Angular Concentration-biased Sampling" (using half the data) need more nuanced discussion regarding the broader implications of potentially excluding certain data points.

3.	The comparison with existing curriculum learning methods should be broader and more deeply analyze GAIN-RL's advantages over a wider range of strategies.

4.	The paper contains several minor typos and inconsistent formatting details that, while not severely impacting understanding, should be addressed for improved professionalism.

---

> ### Author Rebuttal · Authors · 2025-07-31
>
> We thank the reviewer for the very positive rating and insightful feedback. We are encouraged to hear that the reviewer finds our discussion of leveraging angle concentration as a model-inherent learning signal to be theoretically sound and empirically compelling, and considers our proposed approaches to be both original and highly significant for improving the efficiency of RL fine-tuning. We hope the following discussions can address the reviewer's remaining questions.
>
>
>
> > ***Weakness1. The claim of "no additional cost" for computing accuracy and angle concentration for probability updates requires more detailed justification across various training setups to ensure consistent fairness.***
>
> **Responce**: We sincerely thank the reviewer for the thoughtful comment and fully agree that the claim of “no additional cost” should be carefully justified across different training setups. We appreciate the opportunity to clarify this point.
>
> In our implementation, both **accuracy** and **angle concentration** are collected without introducing any additional inference steps. Specifically, accuracy is computed as part of the loss evaluation for each training sample, which is already standard in most RL fine-tuning frameworks. Angle concentration, on the other hand, is derived from the logits during the forward pass, which the model performs anyway. Therefore, **both signals are gathered naturally during training without any extra computational overhead.**
>
> **Importantly, this process is independent of model size, architecture, or hyperparameter settings, and applies consistently across different training setup.** We will revise the submission to make this explanation clearer and more explicit.
>
> We sincerely appreciate the reviewer’s careful reading and valuable feedback, which helps us improve the clarity and fairness of our claims.
>
>
>
> > ***Weakness2. The paper's strong results with "High Angular Concentration-biased Sampling" (using half the data) need more nuanced discussion regarding the broader implications of potentially excluding certain data points.***
>
> **Responce**: We sincerely thank the reviewer for this thoughtful and insightful comment. As the reviewer rightly pointed out, Figure 8 deserves a more nuanced explanation. We fully agree and are happy to provide a deeper analysis of this promising result.
>
> Figure 8 illustrates the effect of our proposed High Angular Concentration-biased Sampling method, which selectively excludes samples with low angular concentration. The results suggest that **removing such samples can significantly improve training efficiency while preserving gradient effectiveness and stability.** This observation is supported by several layers of evidence:
>
> 1. **Theoretical justification via Equation (4):** Low angular concentration samples reduce the mean and increase the variance of training gradients. As shown in Equation (4), angular concentration among hidden states directly influences the Frobenius norm of the gradient through the sum of cos θ terms. A higher angular concentration yields stronger, more consistent gradient updates, accelerating the learning process and improving efficiency.
> 2. **Empirical support from the neuron activation patterns (Figure 6):** As shown in Figure 6, high-angular-concentration samples tend to activate a focused cluster of core neurons, enabling effective gradient accumulation and rapid formation of high-performing neuron clusters. In contrast, low-concentration samples activate more scattered neurons, leading to dispersed gradient directions. Although more samples tend to activate core neurons as training progresses, a subset of low-concentration samples continues to produce noisy and inconsistent activations, thereby reducing overall training stability and efficiency.
> 3. **Alignment with prior research findings:** Several prior works [1,2,3] have shown that not all training samples contribute positively to learning, and removing "negative gain" samples can lead to faster and more stable optimization. Our method provides a practical and computationally efficient way to filter out such low-value data points based on angular concentration.
>
> We will expand our discussion of this result in the revised version and plan to explore this direction further in future work.
>
> Once again, we sincerely appreciate the reviewer’s constructive feedback, which has helped us improve the depth and completeness of our paper.
>
>
>
> > ***Weakness3. The comparison with existing curriculum learning methods should be broader and more deeply analyze GAIN-RL's advantages over a wider range of strategies.***
>
> **Responce**:  We sincerely thank the reviewer for this valuable suggestion. We fully agree that a broader and more in-depth comparison with existing curriculum learning and data manipulation strategies would strengthen our paper. Below, we elaborate on the key advantages of GAIN-RL over a wider range of prior approaches:
>
> 1. **Model-aware signal vs. static data assumptions**: A core distinction of GAIN-RL lies in its use of the *model’s own perception* of data difficulty, rather than relying solely on static properties of the data distribution. Prior approaches—such as curriculum learning and expert data selection—typically define difficulty based on external metrics or predefined heuristics, which do not account for how the target model perceives different examples. As shown in Figure 2 of the main paper, we observe that different models often perceive the same data samples with very different levels of difficulty. This highlights a limitation of model-agnostic methods. In contrast, GAIN-RL leverages model-internal feedback to dynamically guide data selection, resulting in more efficient and tailored training.
> 2. **High generality and broad applicability**: GAIN-RL is model- and dataset-agnostic, making it easy to integrate into a wide range of training pipelines. Many existing methods are only effective under specific conditions—for instance, expert data filtering (e.g., s1 and limo) requires large, high-quality, and diverse datasets (e.g., s1 collects 59k samples across 16 datasets), while curriculum learning relies on datasets with clear and measurable difficulty variations. These constraints limit their scalability to everyday training scenarios. In contrast, GAIN-RL requires no special assumptions about data quality or model type: it simply utilizes a lightweight pre-filling stage to collect difficulty signals from the model itself, allowing it to be broadly deployed across diverse training settings.
> 3. **Minimal computational and human cost**: GAIN-RL can be implemented with only minute-level GPU overhead, whereas many prior methods require substantial resources and human or agent-based evaluation. For example, s1 uses Claude 3.5 to score samples by difficulty and diversity, and curriculum learning strategies often depend on human performance statistics from competitions to assess question difficulty. These approaches are computationally intensive and prohibitively costly for most researchers. GAIN-RL, on the other hand, relies solely on the model's own forward pass, making it far more accessible and scalable for real-world use.
>
> While we included several representative baselines in our current experiments, their scope was limited due to the aforementioned challenges of reproducibility and cost. For instance, curriculum-based difficulty metrics are mostly defined on math datasets and do not generalize well to domains like code. Similarly, expert filtering approaches depend on diverse and curated datasets, along with expensive agent assistance, which makes them difficult to reimplement in general-purpose training scenarios.
>
> We are actively collecting additional baselines and will incorporate broader experimental comparisons and deeper discussions of GAIN-RL's advantages in the revised version. We once again thank the reviewer for this constructive feedback, which will help us significantly improve the completeness and rigor of our work.
>
>
>
> > ***Weakness4. The paper contains several minor typos and inconsistent formatting details that, while not severely impacting understanding, should be addressed for improved professionalism.***
>
> **Responce**:  We sincerely thank the reviewer for carefully reading our paper and pointing out the formatting and typographical issues. We will thoroughly proofread the manuscript and correct these minor errors in the revised version to improve clarity and overall professionalism.
>
>
>
> **References:**
>
>  *[1] Schaul T, Quan J, Antonoglou I, et al. Prioritized experience replay[J]. arXiv preprint arXiv:1511.05952, 2015.*
>
>  *[2] Pleiss L S, Sutter T, Schiffer M. Reliability-Adjusted Prioritized Experience Replay[J]. arXiv preprint arXiv:2506.18482, 2025.*
>
>  *[3] Isele D, Cosgun A. Selective experience replay for lifelong learning[C]//Proceedings of the AAAI conference on artificial intelligence. 2018, 32(1).*

---

### Decision · Program_Chairs · 2025-09-17

**Decision:**

Accept (spotlight)

**Comment:**

The paper proposes GAIN-RL, a reinforcement learning fine-tuning (RFT) algorithm for LLM  that improves sample efficiency and reduces computational cost. The key idea is to use angle concentration—cosine similarity among hidden states—as a model-intrinsic signal to guide data reordering and dynamic sampling. By leveraging this signal with Gaussian-based sampling updates, GAIN-RL achieves faster training and stronger performance compared to vanilla GRPO and related baselines.

All three reviewers rated the paper Accept (5), and they agree on the technical soundness, originality, and strong empirical validation. While some questions remain (e.g., broader baselines, clearer theoretical exposition), the consensus is that the contribution is novel and timely. Therefore, I recommend to accept the paper.